# Research

mathematical modelling/theoretical biology/health and disease and epidemiology

malaria, mosquito, within-host model, Bayesian inference, extrinsic incubation period

**Author for correspondence:**
Olivia F. Prosper
e-mail: oprosper@utk.edu

[†]These authors contributed equally.

# The impact of within-vector parasite development on the extrinsic incubation period

Lauren M. Childs[1,†] and Olivia F. Prosper[2,†]

[1]Department of Mathematics, Virginia Tech, 225 Stanger St, Blacksburg, VA 24060, USA
[2]Department of Mathematics, University of Tennessee, 1403 Circle Dr, Knoxville, TN 37996, USA

OFP, 0000-0002-8042-0616

Mosquito-borne diseases, in particular malaria, have a significant burden worldwide leading to nearly half a million deaths each year. The malaria parasite requires a vertebrate host, such as a human, and a vector host, the *Anopheles* mosquito, to complete its full life cycle. Here, we focus on the parasite dynamics within the vector to examine the first appearance of sporozoites in the salivary glands, which indicates a first time of infectiousness of mosquitoes. The timing of this period of pathogen development in the mosquito until transmissibility, known as the extrinsic incubation period, remains poorly understood. We develop compartmental models of within-mosquito parasite dynamics fitted with experimental data on oocyst and sporozoite counts. We find that only a fraction of oocysts burst to release sporozoites and bursting must be delayed either via a time-dependent function or a gamma-distributed set of compartments. We use Bayesian inference to estimate distributions of parameters and determine that bursting rate is a key epidemiological parameter. A better understanding of the factors impacting the extrinsic incubation period will aid in the development of interventions to slow or stop the spread of malaria.

## 1. Introduction

Mosquito-borne diseases such as malaria, dengue and Zika have a significant burden worldwide with over 50% of the world's population in areas with risk of transmission [1]. As these diseases utilize a mosquito host for part of their life cycle, it is essential to consider dynamics of the pathogen in the mosquito. A key quantity in understanding the spread of these diseases involves the time period of pathogen development in the mosquito until transmissibility. This period, known as the extrinsic incubation period (EIP), from acquired infection in the mosquito until the ability to pass on the infection to another host, remains poorly understood [2].

EIP is difficult to quantify in part because many factors influence its outcome, and accounting for these factors is experimentally difficult. Here, we focus on EIP of malaria in individual *Anopheles* mosquitoes. Typically, malaria parasites enter the mosquito via a blood meal from an infectious vertebrate. Although asexual parasites may be present in the blood meal, only the sexually dimorphized mature gametocytes will be able to survive in the environmental conditions in the mosquito [3]. Once in the mosquito, gametocytes undergo gametogenesis to form male and female gametes. These haploid male and female gametes fuse to form diploid zygotes, which develop into ookinetes. The ookinetes traverse the lining of the midgut and develop into tetraploid oocysts [4]. This all happens on a time scale of minutes to a day [5]. Then, oocysts remain attached to the midgut wall while the parasites themselves multiply repeatedly within them [6,7]. After about 10–11 rounds of endomitosis, the oocysts burst, leading to release of parasites in the sporozoite form [8]. The time period from oocyst formation to oocyst bursting differs by species with *Plasmodium falciparum* bursting after as few as 10 days [9], while *Plasmodium berghei* burst closer to 14 days [10]. Once burst from the oocyst, sporozoites must transit to the salivary glands before the mosquito is considered infectious. Although individual oocysts may burst over a series of days, the arrival of the first sporozoites in the salivary glands heralds the ending of the EIP.

Many factors probably influence the length of the EIP including parasite subspecies, mosquito species, initial densities and environmental conditions (see [11–13] for examples of estimating EIP duration under different experimental conditions). To estimate EIP duration and to examine the effect of initial parasite densities on EIP, we considered data resulting from a series of experiments measuring temporal oocyst and sporozoite densities of *Plasmodium berghei* in *Anopheles stephensi* mosquitoes reported in the doctoral thesis [14]. While the results of several related experiments are published in [14–17], along with some preliminary temporal oocyst count data in [18], the data extracted for our manuscript remain unpublished. These unpublished data are especially informative in that the authors reduce variability by starting infections at the ookinete stage rather than with gametocytes in the blood. This reduces fluctuations arising from gamete formation and mating. We draw upon this data to examine the role of parasite life stage parameters on the EIP.

Mathematical models have been an important tool in understanding malaria dynamics both at the epidemiological level, e.g. [19–24] and references within, and the within-host level [5,25–29]. However, only a limited set of mathematical work has focused on understanding parasite dynamics in the mosquito, including [5,14,15,18,25,29], with some of the models remaining unpublished [14]. While the models in [14], which fit to the same data we consider, provide insight into parasite dynamics within the mosquito, these models are not intended to examine the EIP and, thus, are not able to reproduce the oocysts and sporozoite dynamics simultaneously. Here, we build upon these works, incorporating experimental data from [14] on oocyst counts and sporozoite score from particular ookinete starting densities. These data allow us to examine in detail the dynamics in the oocyst phase, the primary driver of the length of the EIP.

We begin in §2 with a discussion of the data. In §3, we introduce two versions of our model of parasite development between the ookinete stage and the sporozoite stage within mosquitoes. In §4, we discuss our choice of parametrization and model selection with regard to various starting ookinete densities. In §5, we assess the implications for EIP and find that intermediate ookinete densities lead to the shortest predicted EIP and that bursting rate has the largest impact on EIP. Finally, in §6, we discuss our findings, and in §7, we summarize the implications.

# 2. Data

In [14], a series of three controlled experiments were conducted to examine the role of ookinete density on counts of subsequent parasite stages (oocyst number and sporozoite score). Specific details of the methodology can be found in [14], chapter 4. Each experiment involved three experimental conditions (different ookinete densities). Three ookinete densities (100, 400 and 2000 µl$^{-1}$) were used for each of experiments 1 and 2. In experiment 3, the ookinete densities (50, 250 and 1000 µl$^{-1}$) were altered to explore additional initial conditions [14]. In short, for each experiment, approximately 1500–2000 mosquitoes were fed on blood containing various ookinete densities. Ookinete blood was formed by taking gametocytaemic blood from infected mice and incubating for 24 hours, before processing to form the defined ookinete levels. Following blood feeding, 20 mosquitoes were dissected every day for the first 6 days and every 1–2 days after that until the end of each experiment, which varied in length with the longest being 42 days. Within each dissected mosquito, the number of oocysts were counted, and the sporozoites in the salivary glands were assigned a score.

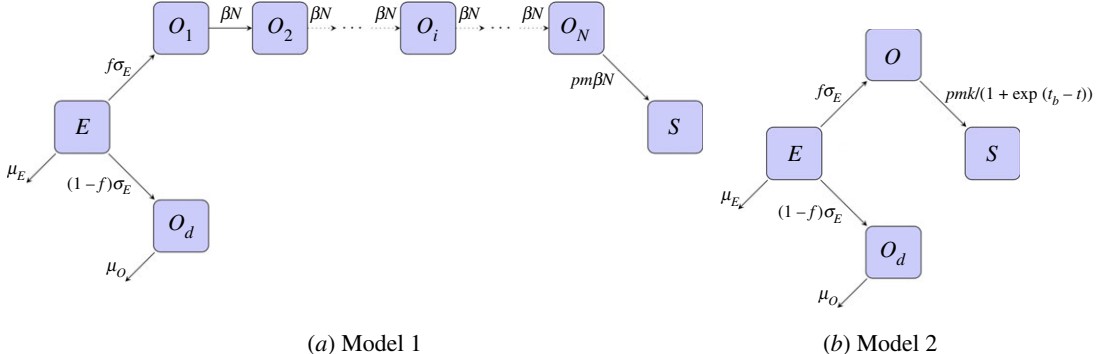

**Figure 1.** Model schematics. Our models consist of ookinetes ($E$), bursting oocysts ($O$), non-bursting oocysts ($O_d$) and sporozoites ($S$) compartments. Ookinetes die at rate $\mu_E$ and transition to oocysts at rate $\sigma_E$ with $f$ fraction ultimately bursting. Non-bursting oocysts die at rate $\mu_O$. (a) Model 1 assumes gamma-distributed bursting with shape parameter $N$ and scale parameter $1/(\beta N)$ and $N$ bursting oocyst compartments. (b) Model 2 assumes time-dependent bursting at the rate $k/(1 + \exp{(t_b - t)})$. Upon bursting, $m$ sporozoites burst from each oocyst and reach the salivary gland with probability $p$.

Two microlitres of blood were considered to be the average size of a mosquito blood meal. Thus, initial starting ookinete numbers were approximately double the densities reported above.

Sporozoites were not directly counted; instead, sporozoite abundance levels were assigned a score between 0 and 4 on a log scale. A score of 0 indicated an absence of sporozoites. A score of 1 indicated 1 to 10 sporozoites; a score of 2 meant 11 to 100 sporozoites; a score of 3 meant 101 to 1000 sporozoites; and a score of 4 meant 1001 to 10 000 sporozoites. No record of sporozoites above 10 000 was indicated [14].

We extracted the data on mean oocyst number (figure 6.14 in [14]) and mean sporozoite score (figure 6.16 in [14]) using Plotdigitzer version 4.2 [30] and GRABIT version 1.0.0.1 [31]. Raw data (not means) can be found in figures 6.2 and 6.3 within [14], but as the points overlap, we could not accurately digitize this data. Only sporozoite scores and prevalence, not counts, were reported in [14]. To account for manual error during the digitization process, we rounded all times to integer days post engorgement and rounded oocyst counts to whole numbers. We rounded sporozoite scores to nearest tenth decimal place.

In electronic supplementary material, figure S1, we present the data—mean oocyst count and mean sporozoite score—by experiment and condition extracted from [14]. Because experiment 2 is a replicate of experiment 1, we combined these datasets for our analysis and refer to the resulting six datasets (or treatments) by their corresponding initial ookinete number, $E(0)$. The data can also be seen in our main text, figure 3, black dots.

# 3. Model formulation and selection

To focus on the timing of sporozoite arrival in the salivary glands, we build deterministic, ordinary differential equation (ODE) models (autonomous and non-autonomous). In particular, we compare two models, which differ in how they capture the delay in oocyst rupture following oocyst formation. Schematics of the two models are shown in figure 1. In Model 1, the rupturing oocyst stage is divided into $N$ sequential equally long stages, which is analogous to assuming time to oocyst rupture is gamma distributed when $N > 1$, and exponential when $N = 1$ [32]. In Model 2, the delay in oocyst rupture is captured by a time-dependent rupture function. Both models include a single ookinete stage and separate oocysts into bursting oocysts and non-bursting oocysts. We find the separation into two oocyst categories necessary to account for the simultaneous persistence of oocysts and stabilization of sporozoite score. Oocyst counts were positive to the final day counts were performed, past 30 days in all cases and up to day 42 in the longest experiment (electronic supplementary material, figure S1, blue squares). After about 25 days, the sporozoite scores do not appear to increase, indicating that oocysts were no longer bursting (electronic supplementary material, figure S1, red triangles).

Our modelling is motivated by the work of [14]. The author considers two successive model structures with the goal of studying possible density-dependence in the ookinete-to-oocyst transition and oocyst-to-sporozoite transition: (i) multiple ookinete stages to produce a single oocyst stage, and (ii) multiple oocyst stages starting at day 10 to produce sporozoites. Model (i) allows for a gamma-distributed ookinete stage, whereas model (ii) allows for a gamma-distributed oocyst stage. They fit (i) to oocyst data and fit (ii) to sporozoite data. Although we find excellent replicability of their results assuming a similar model structure (not shown), we find that neither of their models alone can fit

both the oocyst and sporozoite data simultaneously. Here, we build two models to account for both oocyst and sporozoite data together, to investigate questions about the EIP duration.

Our two formulations differ in how bursting of oocysts is modelled. Model 1 incorporates multiple, identical oocyst compartments representing a gamma-distributed time until bursting. Model 2 uses a time-dependent bursting function that is sigmoidal in shape. Electronic supplementary material, figure S3, shows the difference in the bursting time of the two models. As $N$ increases, the cumulative distribution function of Model 1 approaches that of Model 2. However, the probability density function (PDF) for Model 1 always remains lower than for Model 2.

## 3.1. Model 1: gamma-distributed bursting time model

In Model 1, we assume a single ookinete stage and multiple oocyst stages for those that eventually burst, but a single stage of non-bursting oocysts. This is equivalent to assuming that the ookinete duration is exponentially distributed, time to oocyst rupture is gamma-distributed, and oocyst mortality is exponentially distributed. This latter fact arises because we only allow for death of oocysts in the population of non-bursting oocysts. Our Model 1 system of equations for ookinetes ($E$), bursting oocysts of stage $i$ ($O_i$), non-bursting oocysts ($O_d$), and sporozoites ($S$) is

$$\frac{dE}{dt} = -\mu_E E - \sigma_E E,$$

$$\frac{dO_1}{dt} = f\sigma_E E - \beta N O_1,$$

$$\frac{dO_i}{dt} = \beta N O_{i-1} - \beta N O_i, \quad i = 2, \ldots, N,$$

$$\frac{dO_d}{dt} = (1-f)\sigma_E E - \mu_O O_d,$$

$$\frac{dS}{dt} = pm\beta N O_N,$$

where $\mu_E$ is the mortality rate of ookinetes, $\sigma_E$ is the transition rate of ookinetes to oocysts, $f$ is the proportion of transitioning ookinetes that become bursting oocysts, $\mu_O$ is the mortality rate of non-bursting oocysts, $\beta N$ is the transition rate between the $N$ oocyst stages, $m$ is the number of sporozoites that burst from an oocyst, and $p$ is the probability of sporozoites reaching the salivary glands. See table 1 for a list of the parameters.

## 3.2. Model 2: time-dependent bursting model

In Model 2, we assume single stages for ookinetes, bursting oocysts and non-bursting oocysts. However, we assume that bursting oocysts follow a time-dependent bursting function as in [25], similar to the step function employed in [5,29]. Our Model 2 system of equations for ookinetes ($E$), bursting oocysts ($O$), non-bursting oocysts ($O_d$) and sporozoites ($S$) is

$$\frac{dE}{dt} = -\mu_E E - \sigma_E E,$$

$$\frac{dO}{dt} = f\sigma_E E - \frac{k}{1 + \exp(t_b - t)}O,$$

$$\frac{dO_d}{dt} = (1-f)\sigma_E E - \mu_O O_d,$$

$$\frac{dS}{dt} = pm\frac{k}{1 + \exp(t_b - t)}O,$$

where $\mu_E$, $\sigma_E$, $f$, $\mu_O$, $m$ and $p$ are as in Model 1. Here, $\frac{k}{1+\exp(t_b-t)}$ is the time-dependent bursting rate of oocysts with $k$ being the maximal rate and $t_b$ being the time when bursting occurs at half the maximal rate. See table 1 for a full description of parameters, compared between models. See §4.3 for a discussion on how $k$ and $t_b$ relate to EIP.

# 4. Parameter estimation

In order to examine these models and their parameters for implications on the EIP, we fit the models to the data described above and found in [14]. We do not fit parameters $m$ and $p$, which together determine

**Table 1.** Parameter descriptions.

| Symbol | Description | Model | Bounds |
|---|---|---|---|
| $\mu_E$ | mortality rate of ookinetes | 1,2 | $\mathcal{U}(0, 2)$ |
| $\sigma_E$ | transition rate of ookinetes to oocysts | 1,2 | $\mathcal{U}(0, 2)$ |
| $\mu_O$ | mortality rate of non-bursting oocysts | 1,2 | $\mathcal{U}(0, 2)$ |
| $f$ | proportion of transitioning ookinetes that become bursting oocysts | 1,2 | $\mathcal{U}(0, 1)$ |
| $1/\beta$ | average duration of bursting oocyst stage | 1 | $\mathcal{U}(0, 10)$ |
| $k$ | maximal bursting rate | 2 | $\mathcal{U}(0, 10)$ |
| $t_b$ | time when bursting occurs at half the maximal rate | 2 | $\mathcal{U}(0, 42)$ |
| $p$ | probability of sporozoites reaching the salivary glands | 1,2 | – |
| $m$ | number of sporozoites that burst from an oocyst | 1,2 | – |

the number of sporozoites per oocyst that reach the salivary glands. Previous work shows a range of 54–72 sporozoites per oocyst reach the salivary glands [17]. As $p$ and $m$ always appear as a product, we choose $pm = 60$.

## 4.1. Likelihood function

We fit both models to a combination of the oocyst and sporozoite score data. We use the Poisson likelihood for the oocysts

$$\mathcal{L}_{\mathcal{O}}(o|\lambda_O) = \prod_{i=1}^{M_O} \frac{\lambda_O^{o_i}(t_i)\, e^{-\lambda_O(t_i)}}{o_i!},$$

where $o_i$ is the data of mean total oocysts (combining bursting and non-bursting) at time point $t_i$, $\lambda_O(t_i) = O_d(t_i) + \sum_{j=1}^{N} O_j(t_i)$ is the model output of the total oocysts at time point $i$ and $M_O$ is the total number of time points in the oocyst data.

Likewise, we use the Poisson likelihood for the sporozoite number

$$\mathcal{L}_{\mathcal{S}}(s|\lambda_S) = \prod_{i=1}^{M_S} \frac{\lambda_S^{s_i}(t_i)\, e^{-\lambda_S(t_i)}}{s_i!},$$

where $s_i$ is the data of sporozoite count (calculated by raising 10 to the power equal to the sporozoite score) at time point $t_i$, $\lambda_S(t_i) = S(t_i)$ is the model output of the total sporozoite number at time point $t_i$, and $M_S$ is the total number of time points in the sporozoite data. We combine the oocyst and sporozoite likelihoods such that the negative log-likelihood (NLL) is

$$\text{NLL} = -\ln(\mathcal{L}_{\mathcal{S}}) - \ln(\mathcal{L}_{\mathcal{O}})$$

$$= -\sum_{i=1}^{M_O} \left( o_i \ln(\lambda_O(t_i)) - \lambda_O(t_i) - \sum_{j=1}^{i} \ln(o_i) \right)$$

$$- \sum_{i=1}^{M_S} \left( s_i \ln(\lambda_S(t_i)) - \lambda_S(t_i) - \sum_{j=1}^{i} \ln(s_i) \right).$$

For the model output, we add a small value, $\varepsilon = 0.01$, to the mean scores to ensure non-zero values enter the likelihood [33].

We employ this likelihood function in two fitting methods, multistart optimization and Markov chain Monte Carlo (MCMC), described below. The former allows us to explore the parameter space of both models, but does not provide information on parameter variability or correlations. The latter allows us to explore both of these aspects (parameter variability and correlations) but is more computationally expensive, particularly for large ODE systems, so we restrict this methodology to Model 2.

Our model differs from the data in how sporozoites are quantified. In the data, sporozoite score is a measurement of the abundance of sporozoites on a log scale, while we track the calculated density

directly in the model. For comparison, we transform the sporozoite score to sporozoite number by raising 10 to the reported average value, i.e. the 'anti-log'. As noted in the original presentation of the data [14], there are other ways to transform the sporozoite score such as the arithmetic or geometric mean. We examine the effect of other choices for transforming the sporozoite score in the electronic supplementary material, figure S2 and find that our results for EIP are not sensitive to the method for transforming the score. The 'anti-log' transformation consistently overestimates the true value. However, it is unclear if there is over- or under-estimation with other methods, so we present results using 10 raised to the average score.

## 4.2. Parameter bounds

We fit five parameters for Model 1 ($\mu_E$, $\sigma_E$, $\mu_O$, $f$, $\beta$) for varying numbers of oocyst compartments, $N$, and fit six parameters for Model 2 ($\mu_E$, $\sigma_E$, $\mu_O$, $f$, $k$, $t_b$). See table 1 for a full set of parameter definitions and how they are used in the models. We bound the region of optimization for our parameters based on biological knowledge, discussed below.

There is evidence that the rate of leaving the ookinete compartment is constrained such that $\mu_E + \sigma_E \leq 2$, indicating an average time of 0.5 days as an ookinete [29,34]. However, this constraint is observed for *P. falciparum*, not *P. berghei*, the parasite studied in the data presented here. Thus, as the ookinete stage is known to be short, we bound $\mu_E$ and $\sigma_E$ separately between 0 and 2. As oocysts persist from days to weeks, we expect the mortality rate, equivalent to one over the average time as a non-bursting oocyst, to be quite small. For consistency with mortality of the ookinete stage, $\mu_E$, we bound $\mu_O$ between 0 and 2.

The idea of two oocyst sub-populations, one which bursts and one which does not, has not, to our knowledge, been modelled before. However, consistent with observations made in [14], given both the persistence of oocysts for a long period (up to the maximum 42 days) and lack of the additional appearance of sporozoites after about day 25, we find the presence of two separate types a consistent way to account for this observation. However, given the lack of prior discussion on this, we allow $f$, which is the proportion of oocysts that burst, to vary over its full range between 0 and 1.

As expected from the form of the bursting equation in Model 2, we find that $k$ and $t_b$ are correlated and bound them accordingly. To understand their correlation, we examine a population, $x$, only subject to bursting described by

$$\frac{\mathrm{d}x}{\mathrm{d}t} = \frac{-k}{1 + \exp(t_b - t)}x,$$

which can be solved explicitly for $x(t)$ in terms of $t_b$, $k$ and the initial population, $x(0) = x_0$, as

$$x(t) = x_0 \left(\frac{1 + e^{t_b}}{e^t + e^{t_b}}\right)^k.$$

We find the time, $t^*$, when the burst population is half the size of the initial population as

$$t^* = \ln(2^{1/k}(1 + e^{t_b}) - e^{t_b}). \tag{4.1}$$

This quantity, $t^*$, will form a component in our computation of EIP for Model 2. There is clear interdependence on $k$ and $t_b$ (electronic supplementary material, figure S4). Rapid decreases in $t^*$ are seen with increases in $k$ when $k < 0.1$, but only a slow decrease in $t^*$ with $k$ when $k > 0.1$. For $t_b$, there are roughly linear increases in $t^*$ as $t_b$ increases. In electronic supplementary material, figure S4, we show the level curves of $t^*$ as $k$ and $t_b$ are varied. When $k = 1$, we find that $t^* \approx t_b$ with the approximation getting better as $t_b$ increases. We consider $t_b$ across all time values of the data, $t_b \in [0, 42]$. As a result of our analysis of $t^*$ found above, we restricted $k$ to be between 0 and 10.

## 4.3. Extrinsic incubation period

We estimated the extrinsic incubation period (EIP) using the formula

$$\mathrm{EIP} = \frac{1}{\sigma_E + \mu_E} + t^*, \tag{4.2}$$

where for Model 1, $t^*$ is the time at which the cumulative distribution function for the gamma distribution with shape $N$ and scale $1/(\beta N)$ equals 0.5 (i.e. the time at which the PDF attains its median value), and for Model 2, $t^*$ is defined as in equation (4.1) (the time at which half of the initial oocysts have ruptured). The first term, $1/(\sigma_E + \mu_E)$, is the average duration of the ookinete stage. Note that the epidemiological definition for EIP, the time from gametocyte ingestion until emergence of sporozoites in the salivary glands, is slightly different. Our two approaches for calculating $t^*$ lead to the most parsimonious comparison of EIP across models.

## 4.4. Parameter fitting: multistart

The likelihood function may have many local minima. To obtain initial point estimates for the parameters for Model 1, $\sigma_E$, $\mu_E$, $\mu_O$, $f$ and $\beta$, and for Model 2, $\sigma_E$, $\mu_E$, $\mu_O$, $f$, $t_b$ and $k$, we implement the multistart routine within the Matlab Global Optimization Toolbox, using 10 starting points for the initial vector of parameter values. The function fmincon was used to find a local minimum for each starting point. For each starting parameter set, we fit Model 1 for $N \in \{2, 3, 10, 20, 30, 40, 50, 75, 100\}$. Likewise, we fit Model 2 ten times, once for each starting parameter set. This procedure resulted in up to 10 unique parameter estimates, where each parameter set corresponds to a local minimum. The exact number of unique parameter sets depended on whether the optimization routine converged or not for a particular starting point and whether those that converged to different minima. Convergence was determined by the exit flag in the algorithm with a flag of 1 or 2 indicating at least one incidence of convergence. See electronic supplementary material, table S1, for records of our convergence. For each model, we chose the parameter set resulting in the smallest local minimum as our best estimate. The first initial parameter vector in the multistart routine was chosen to be $\{\sigma_E, \mu_E, \mu_O, \beta, f\} = \{0.14, 1.85, 0.045, 0.057, 0.045\}$ for Model 1 and $\{\sigma_E, \mu_E, \mu_O, k, f, t_b\} = \{0.14, 1.85, 0.045, 0.057, 0.045, 10\}$ for Model 2. The remaining nine starting parameter sets are automatically selected at random within the prescribed bounds. The parameter bounds used in multistart are identical to the range of priors found in table 1.

### 4.4.1. Model selection

We choose between the proposed models using a modification of the Akaike information criteria (AIC). AIC uses the likelihood of the model in the context of the number of parameters necessary, penalizing more complicated models [35]. The AIC is given by

$$\text{AIC} = 2D - 2\ln(\mathcal{L}),$$

where $D$ is the number of parameters and $\mathcal{L}$ is the likelihood, described above.

We instead consider the AIC with correction (AICc), which is more appropriate for small sample sizes assuming normality, as it penalizes more complicated models more severely [35]. The AICc is given by

$$\text{AICc} = \text{AIC} + \frac{2D^2 + 2D}{M - D - 1},$$

where $D$ and AIC are as above, and $M$ is the sample size, i.e. the number of data points.

## 4.5. Parameter fitting: Markov chain Monte Carlo

Following our original parameter sweep using multistart, described above in §4.4, we fit Model 2 to the oocyst count and sporozoite score data using MCMC. We use the debInfer package in R [33,36–38], which is designed for Bayesian inference of dynamical models. It uses a random-walk Metropolis–Hastings algorithm [33]. We chose uninformative priors as specified in table 1. Discussion of choices of the parameter bounds is found in §4.2. We use an asymmetric uniform proposal distribution to ensure that all parameters are positive as given by $\mathcal{U}((a/b)\theta, (b/a)\theta)$. The parameter set is given by $\theta$, and we chose $a = 3$ and $b = 4$. Our choices of $a$ and $b$ provide a narrow proposal, which reduces our acceptance rate in certain situations. To compensate for this, we run five chains of at least 100 000 steps for each of the initial oocyst numbers. We choose our starting parameters for the chains using Latin-hypercube sampling with evenly spaced bins across the priors to ensure our starting parameter sets are sufficiently separated. We use the Gelman–Rubin convergence diagnostic [39,40] to confirm our chains have converged. All chains show point estimates of the potential scale reduction factor of 1.05 or less and an upper confidence limit of 1.10 or less (electronic supplementary material, table S2).

## 4.6. Parameter sensitivity analysis

We performed extended Fourier amplitude sensitivity test (eFAST) [41] to determine the impact of our parameters on the fixed output, EIP, and temporally varying output such as oocyst and sporozoite numbers. eFAST can account for non-monotonic relationships between parameters and output [41,42]. In addition, it can give a measure of the total sensitivity to a particular parameter. The first-order sensitivity for parameter $i$, given by $S_i$, is computed from the variance from parameter $i$, denoted by $s_i^2$, divided by the total variance, $s_{\text{total}}^2$

$$S_i = \frac{s_i^2}{s_{\text{total}}^2}.$$

Variance is calculated from the Fourier coefficients at the frequency $j$ of interest by $s_i^2 = 2(A_j^2 + B_j^2)$, where $A_j = \frac{1}{\pi}\int_{-\pi}^{\pi} f(x)\cos(jx)\,\mathrm{d}x$ and $B_j = \frac{1}{\pi}\int_{-\pi}^{\pi} f(x)\sin(jx)\,\mathrm{d}x$ are the Fourier coefficients. The total sensitivity index, given by $S_{Ti}$ for parameter $i$, is found by summing the sensitivity index of all parameters except $i$, given by the complementary set $S_{ci}$, and looking at the remaining variance: $S_{Ti} = 1 - S_{ci}$. We perform eFAST using code originally developed by the Kirschner lab [43].

# 5. Results

## 5.1. Model structure

### 5.1.1. Characteristics required to capture oocyst and sporozoite data simultaneously

Previous modelling efforts have fit the oocyst data and the sporozoite data independently in order to study the transitions between consecutive stages of the sporogonic cycle and explore the possibility that these transitions are density-dependent [14]. In particular, an ookinete-to-oocyst model was fit to oocyst data only and an oocyst-to-sporozoite model was fit to the sporozoite data alone. These two models cannot be used to study our questions about the duration of the EIP and possible dependence of EIP on initial ookinete density. In the case of the ookinete-to-oocyst model, there is no sporozoite compartment to fit to the sporozoite data; whereas the oocyst-to-sporozoite model cannot provide a suitable fit to both oocyst and sporozoite datasets simultaneously (not shown). A major reason for this is that the sporozoite score data rise rapidly following 18 days post infection and plateau by 25 days. With regard to oocyst counts, even out to 42 days, the longest time point measured, there are still oocysts present. These oocysts are falling rapidly between 10 and 42 days, but only the initial portion of this can be explained by the bursting of oocysts. In other words, the previous models either capture the timing and level of the oocyst peak accurately or capture the timing and level of the sporozoite population, but not both.

A parsimonious model change that allows for both of these features simultaneously is to split the oocyst compartment into two populations: one that bursts and one that does not. The bursting oocysts form sporozoites while the non-bursting oocysts merely die off over time. This allows for both a sudden increase in the sporozoite population around day 20 and a significant, but much slower, decrease in the oocyst counts from the peak until the end of the experiments.

Both models introduced here are able to capture these features of the data. For Model 1, a larger number of compartments is often needed to delay the formation of sporozoites. For Model 2, the time-dependent bursting keeps the rate of transition to sporozoites low until near the time of $t_b$. Electronic supplementary material, figure S5, demonstrates our fits for our best parameter combinations using multistart. Best fit parameters for both models using multistart as well as the median values of the posterior distributions of MCMC are found in table 2.

### 5.1.2. Comparison of model with gamma-distributed bursting (Model 1) to model with time-dependent bursting (Model 2)

Across both models and two fitting methods, there is high consistency in fitting results. Figure 2 indicates that, for some $N \in \{2, 3, 10, 20, 30, 40, 50, 75, 100\}$, Model 1 provides a better fit to the data than Model 2 in three of the six scenarios. In particular, for an initial ookinete density $E(0) = 100, 200,$ and 2000, Model 1 with gamma-distributed bursting and an intermediate number of bursting oocyst stages, $N$, produces a

**Table 2.** Fitted parameter values. Best fit parameters for both models using multistart as well as the median values of the posterior distributions of MCMC. The 95% highest density posterior intervals for parameters fit with MCMC can be found in electronic supplementary material, table S4. The best overall model as determined by multistart is indicated in the last column by a star.

| $E(0)$ | Method | $\sigma_E$ | $\mu_E$ | $\mu_O$ | $\beta$ | $f$ | $t_b$ | $k$ | $t^*$ | Model |
|---|---|---|---|---|---|---|---|---|---|---|
| 100 | | | | | | | | | | |
| | Multi | 0.33 | 0.94 | 0.050 | 0.051 | 0.027 | — | — | 19.9 | M 1$^\star$, $N = 30$ |
| | Multi | 0.32 | 0.92 | 0.050 | — | 0.027 | 16.3 | 0.21 | 19.6 | M 2 |
| | MCMC | 0.43 | 1.26 | 0.049 | — | 0.027 | 16.50 | 0.23 | 19.5 | M 2 |
| 200 | | | | | | | | | | |
| | Multi | 0.75 | 0.81 | 0.073 | 0.059 | 0.015 | — | — | 17.2 | M 1$^\star$, $N = 50$ |
| | Multi | 0.74 | 0.80 | 0.073 | — | 0.015 | 16.6 | 0.66 | 17.2 | M 2 |
| | MCMC | 0.88 | 0.98 | 0.072 | — | 0.015 | 16.9 | 0.85 | 17.1 | M 2 |
| 500 | | | | | | | | | | |
| | Multi | 0.23 | 1.44 | 0.039 | 0.061 | 0.051 | — | — | 16.4 | M 1, $N = 50$ |
| | Multi | 0.15 | 0.90 | 0.042 | — | 0.048 | 19.8 | 9.99 | 17.2 | M 2$^\star$ |
| | MCMC | 0.18 | 1.08 | 0.041 | — | 0.049 | 19.5 | 7.23 | 17.2 | M 2 |
| 800 | | | | | | | | | | |
| | Multi | 0.21 | 0.61 | 0.069 | 0.063 | 0.018 | — | — | 16.2 | M 1, $N = 75$ |
| | Multi | 0.23 | 0.66 | 0.069 | — | 0.018 | 17.7 | 1.57 | 17.1 | M 2$^\star$ |
| | MCMC | 0.23 | 0.62 | 0.069 | — | 0.018 | 17.9 | 1.76 | 17.1 | M 2 |
| 2000 | | | | | | | | | | |
| | Multi | 0.14 | 1.85 | 0.045 | 0.058 | 0.044 | — | — | 17.2 | M 1$^\star$, $N = 75$ |
| | Multi | 0.15 | 1.89 | 0.045 | — | 0.044 | 17.3 | 0.88 | 17.5 | M 2 |
| | MCMC | 0.14 | 1.78 | 0.046 | — | 0.043 | 17.4 | 0.93 | 17.5 | M 2 |
| 4000 | | | | | | | | | | |
| | Multi | 0.14 | 1.69 | 0.060 | 0.055 | 0.034 | — | — | 17.8 | M 1, $N = 75$ |
| | Multi | 0.068 | 0.74 | 0.070 | — | 0.034 | 22.4 | 10.00 | 19.8 | M 2$^\star$ |
| | MCMC | 0.069 | 0.75 | 0.070 | — | 0.034 | 22.4 | 9.66 | 19.8 | M 2 |

better fit to the data than Model 2 with time-dependent bursting. However, for these initial experimental conditions, the AICc value for Model 2 is relatively close to that of Model 1 with an optimal number of bursting oocyst stages. In fact, the fits to both the oocyst and sporozoite score data are nearly identical between the Model 2 MCMC median of posterior draws and the best fit model using multistart (figure 3).

A summary of parameter estimates is provided in table 2; the best overall model as determined by multistart is indicated in the last column by a star ($^\star$). For the gamma-distributed bursting model, Model 1, the number of bursting oocyst stages that yields the best fit increases with initial ookinete density $E(0)$. Again, there is high agreement between parameter estimates for the best fit Model 1 and Model 2, and between the two fitting methods (MCMC and multistart). Using Model 1 and a fixed initial ookinete density $E(0)$, we also obtain estimates for parameters $\mu_O$, $f$ and $\beta$, that are similar in magnitude across the number of rupturing oocyst stages, $N$, provided that the fit to data is reasonable (see electronic supplementary material, figure S6). In other words, as long as the fitted solution visually aligns with the data, the optimal parameters found via fitting for each $N$ are similar across all $N$. There is more variability in the values of $\sigma_E$ and $\mu_E$ across $N$, which is probably a consequence of the two parameters being correlated (electronic supplementary material, table S5); in particular, $\mu_E$ and $\mu_E + \sigma_E$ appear, roughly, to increase with increasing number of rupturing oocyst stages $N$ (see electronic supplementary material, table S5, and figures provided in our Zenodo Repository [44]). This suggests that the average duration of the bursting oocyst stage is relatively stable across initial ookinete densities, despite the fact that the distribution of this random variable changes depending on the initial conditions. In other words, if we assume that the time from oocyst formation to oocyst bursting is gamma distributed, the expectation $1/\beta$ is fairly constant, while the shape parameter changes with initial ookinete density. Recall

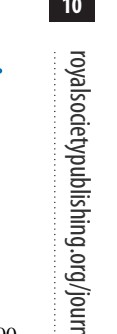

(a)

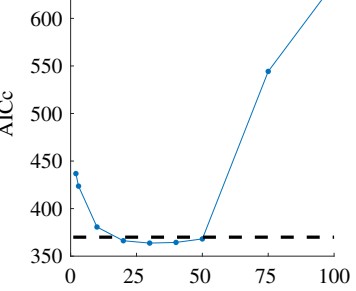

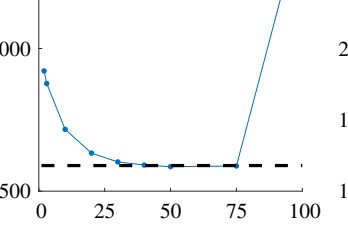

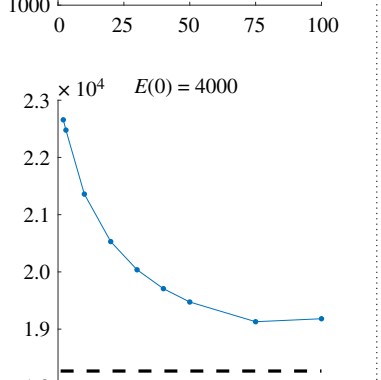

(b)

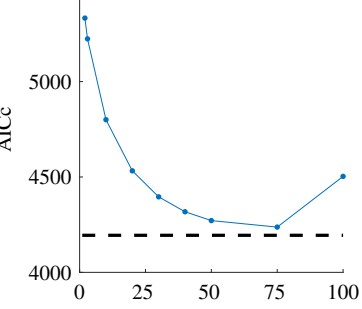

**Figure 2.** Model selection for multistart parameter fitting. AICc plot for each initial ookinete density $E(0)$ for Model 1 with $N$ oocyst stages, where $N \in \{2, 3, 10, 20, 30, 40, 50, 75, 100\}$. The AICc for Model 2 is demonstrated by the dashed horizontal line. AICc values are given in electronic supplementary material, table S3.

that a gamma distribution with mean $1/\beta$ and shape $N$ is equivalent to a gamma distribution with shape $N$ and scale $1/(\beta N)$, which is also the duration of each stage $O_i$ in Model 1.

## 5.2. Oocyst count and sporozoite score with increasing ookinete density

As the initial ookinete density increases, the estimated peaks of oocyst density generally increase and are shifted to later time points (see figure 4), although this is not always the case, as the peaks for 200 and 500 ookinetes are out of order, and similarly for 800 and 2000 ookinetes. While fitting only the ookinete-to-oocyst transition, the results in [14] show a general increase in peak height and shift to the right in peak position for Experiments 1 and 2. The fitted oocyst peaks for all three conditions in Experiment 3 are earlier than expected in relation to initial conditions in Experiments 1 and 2 (considering peaks from fits in figure 6.14 in [14]). Considering a fixed set of parameters and only varying the initial ookinete number, Dawes observes an increase in peak height and a shift in peak location to the right (shown in figure 6.21 in [14]).

The relationship between initial ookinete number and final sporozoite score, however, is consistent with larger initial values corresponding to higher scores. Thus, interestingly the separation of oocyst levels is not directly reflected in the sporozoite scores. Furthermore, as initial ookinete numbers increase, the variability of the timing of sporozoite appearance narrows and is earlier, except for the 4000 ookinetes $\mu l^{-1}$. This reflects the U-shaped EIP values (figure 5), discussed below.

## 5.3. Extrinsic incubation period varies with initial ookinete density

A U-shaped relationship between initial ookinete number, $E(0)$, and EIP is seen across all fitting methods (see figure 5). The highest calculated EIP values are for the lowest initial ookinete numbers, 100, for both model types. Intermediate ookinete densities show shorter EIPs with the shortest obtained for 500 ookinetes. At high ookinete numbers, i.e. 4000, the calculated EIP values are again larger. In addition, we see the most variation in the calculated EIP values for the lowest initial ookinete number (see figure 5). Interestingly, for higher initial ookinete numbers, the time as an ookinete increases (although not to the same extent for 2000 initial ookinetes) while the calculated $t^*$ is much higher for the lowest

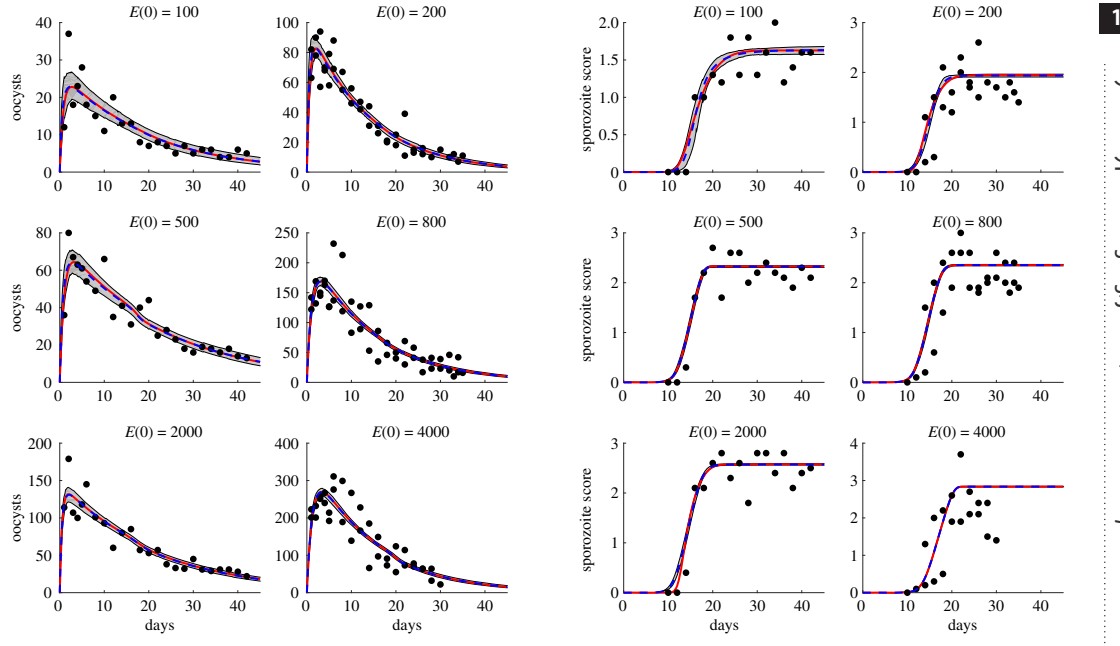

**Figure 3.** Best fit model solutions and data. Comparison of best fit model and data with oocyst count on the left and sporozoite score on the right. The panels are ordered by initial ookinete number, as indicated by their titles. The median of MCMC posterior draws using Model 2 in dashed blue with the 95% highest density posterior interval shown in shaded grey and best fit multistart, which may be Model 1 or Model 2 (see table 2) in solid red. The black points represent data on oocyst count and sporozoite score, extracted from [14].

and highest initial ookinete numbers and relatively similar for intermediate initial ookinete numbers (see electronic supplementary material, figure S7).

## 5.4. Parameter correlations

Focusing on Model 2, which often gives the best fit and is much less computationally expensive, we examined the correlation and variability in parameters using MCMC.

Across all initial conditions we observed strong, yet nonlinear, correlation between the time when bursting occurs at half-maximal rate, $t_b$ and the maximal bursting rate, $k$ (electronic supplementary material, table S5, figures available online at the Zenodo Repository [44]). Similar to the shape of the level curves from electronic supplementary material, figure S4, we observe strong nearly linear increases in $t_b$ with $k$ when $k$ is small. Near $k = 1$, the relationship shifts from large increases to much smaller increases over a short range of $k$ (electronic supplementary material, figure S4). This relationship was expected for a given $t^*$, as shown from our analysis above in §4.2.

We also observed strong, negative correlations between the proportion of transitioning ookinetes that become bursting oocysts, $f$, and the mortality rate of non-bursting oocysts, $\mu_O$ (electronic supplementary material, table S5). This correlation was also expected because the size of the non-bursting oocyst class is dictated by the balance of the input, $(1-f)\sigma_E E(t)$ and, the loss, $\mu_O O_d$. When $f$ is higher fewer oocysts enter the $O_d$ class. Recall that only non-bursting oocysts, $O_d$, are subject to oocyst mortality, $\mu_O$. As some oocysts are present in all experiments at the final time point (42 days for longest experiment) the transition of fewer ookinetes to non-bursting oocysts, i.e. lower $1-f$, necessitates a lower mortality for non-bursting oocysts to persist the entire length of the experiment.

Finally, we observe strong, positive correlations between the mortality rate of ookinetes, $\mu_E$, and the transition rate of ookinetes to oocysts, $\sigma_E$ (electronic supplementary material, table S5). The sum of $\mu_E$ and $\sigma_E$ forms the rate leaving the ookinete class. Consequently, an earlier appearance of oocysts indicates a larger value of this sum. As there is no ookinete data to fit (apart from the initial condition), these values are highly impacted by the levels and timing of oocyst appearance.

In addition, we observed weaker correlations between $f$ and both $\sigma_E$ and $\mu_E$ (electronic supplementary material, table S5). While the correlations with $\mu_E$ were always positive, those with $\sigma_E$ can be both positive and negative. This suggests that the correlations are probably the result of some higher order interaction. We observed weaker, negative correlations between $\mu_O$ and both $\sigma_E$ and $\mu_E$ (electronic supplementary material, table S5). These are probably mediated through the high correlations of $f$ and $\mu_O$.

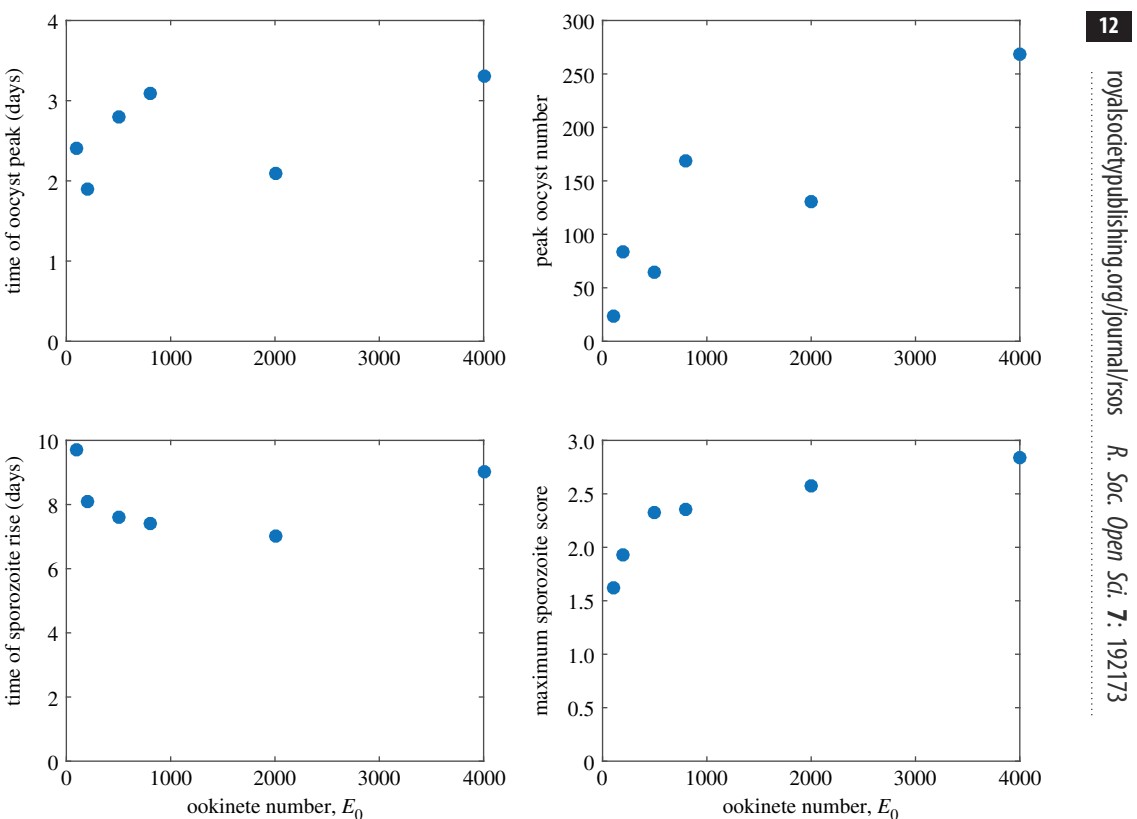

**Figure 4.** Summary metrics of oocyst number and sporozoite score. By initial ookinete number, the time of the oocyst peak (top left), the peak number of oocysts (top right), the first time sporozoite score is above 0.01 (bottom left) and the maximum sporozoite score (bottom right).

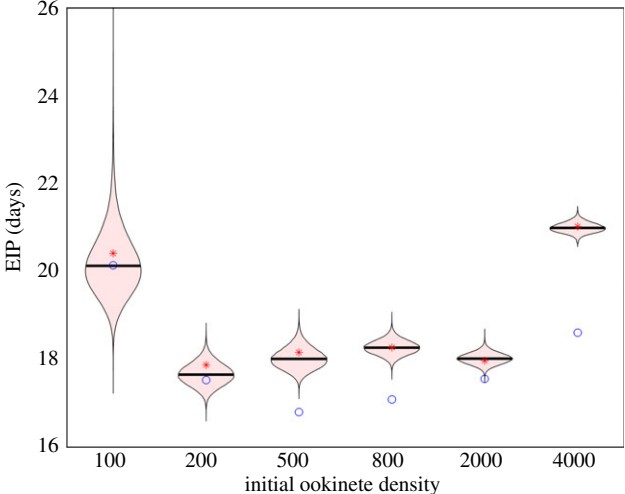

**Figure 5.** Variation in EIP across different initial ookinete numbers. Distribution of EIP with median (black line) for MCMC simulations. Red stars are the EIP from Model 2 fitted with multistart and blue circles are the EIP from Model 1 with optimized $N$ fitted with multistart. All optimized parameters are found in table 2 with 95% highest density posterior intervals found in electronic supplementary material, table S4.

## 5.5. Parameter variability

High variation was found among some parameters (figure 6). We estimated the ookinete-to-oocyst transition rate, $\sigma_E$, in a tight distribution for higher initial ookinete number, i.e. above 200. For $E(0) = 500$, ~800, and 2000, the 95% highest posterior interval (HDPI) for $\sigma_E$ was within 0.1 and 0.3.

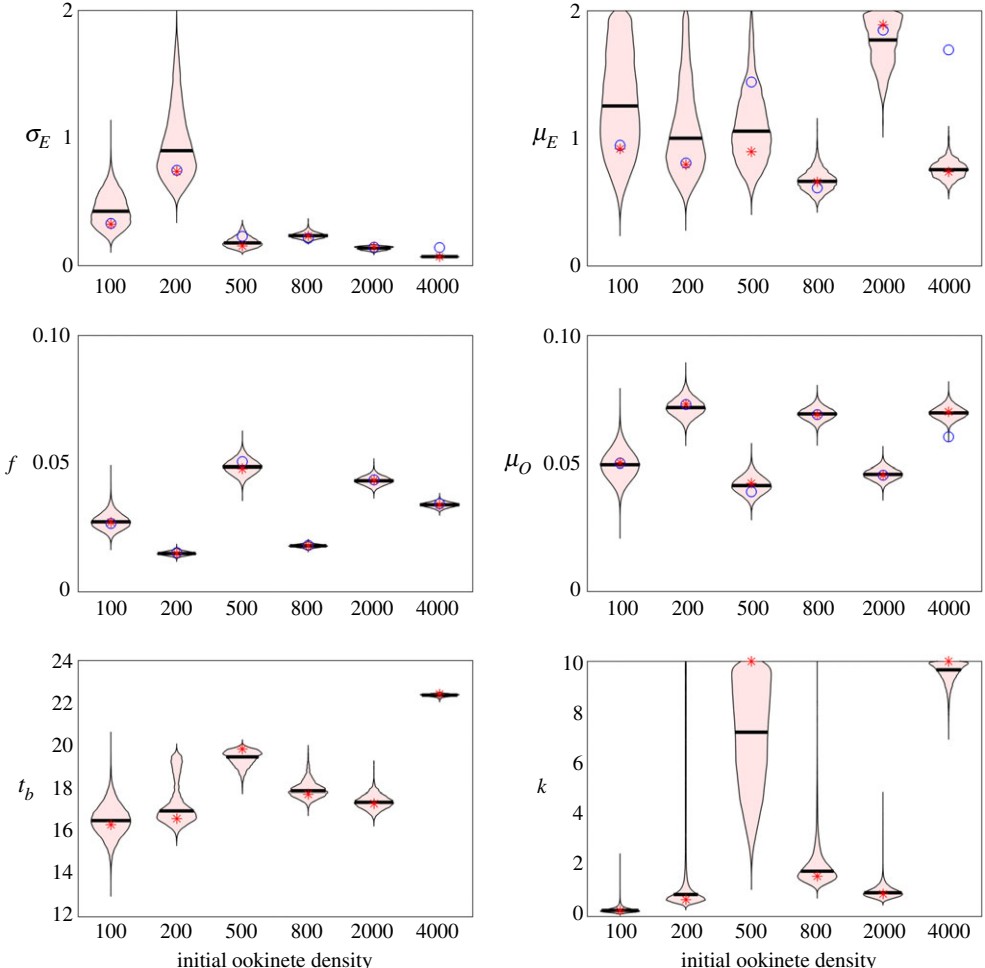

**Figure 6.** Variation in fitted parameters by initial ookinete number. Distribution of posterior density (shaded pink) with median value (black line) for each parameter from Model 2 fit by the MCMC. The prior distributions for each parameter are found in table 1. Optimal parameters fit by multistart are shown as blue circles for Model 1 and red stars for Model 2. Note that in Model 1, there are not the parameters $t_b$ or $k$ so we omit values for Model 1 in the bottom row. The 95% highest density posterior intervals for the MCMC fit parameters are found in electronic supplementary material, table S4.

In contrast, the fits for the ookinete mortality rate, $\mu_E$, are inconsistent across initial ookinete numbers and highly variable, with the 95% HDPI spanning large portions of the prior interval. As described in §5.4, there are interdependencies between the rates associated with entering and persistence in the oocyst classes.

As seen in figure 6, tight fits are found for the fraction of bursting oocysts, $f$, and the oocysts mortality rate, $\mu_O$. In general, the 95% HDPI only spread 0.01 for $f$ and $\mu_O$ (electronic supplementary material, table S4). In fact, values only in a narrow range of the prior (table 1) are recovered (note that the y-axis of figure 6 does not cover the entire prior of $f \in [0, 1]$ and $\mu_O \in [0, 2]$). Furthermore, for any given initial ookinete number, the distribution of these parameters was very narrow. Interestingly, there is a clear visual correlation between experiment and the fits for $f$ and $\mu_O$. Recall that experiments 1 and 2 used initial ookinete densities of 100, 400 and 2000 µl$^{-1}$, while Experiment 3 considered 50, 250 and 1000 µl$^{-1}$. We hypothesize that there may have been some experimental difference between the experiments, e.g. the precision of data collection, the mosquito cohort or any number of environmental conditions. Two differences are noted between experiments in [14]: the age of the cohort of mosquitoes and the researcher performing counts. For experiment 1, the mosquitoes were 4–8 days old at the start of the experiment while for experiments 2 and 3, they were 7–11 days old. Reported oocyst counts and sporozoite score were performed by one researcher for experiments 1 and 2 while they were performed by a different researcher for experiment 3. While statistical analysis was done in [14] to ensure there were no significant differences in the counting, this subtle difference may be appearing in the fits. As this is a re-analysis of data from [14], we are unable to determine the contribution of experimental factors beyond what is reported.

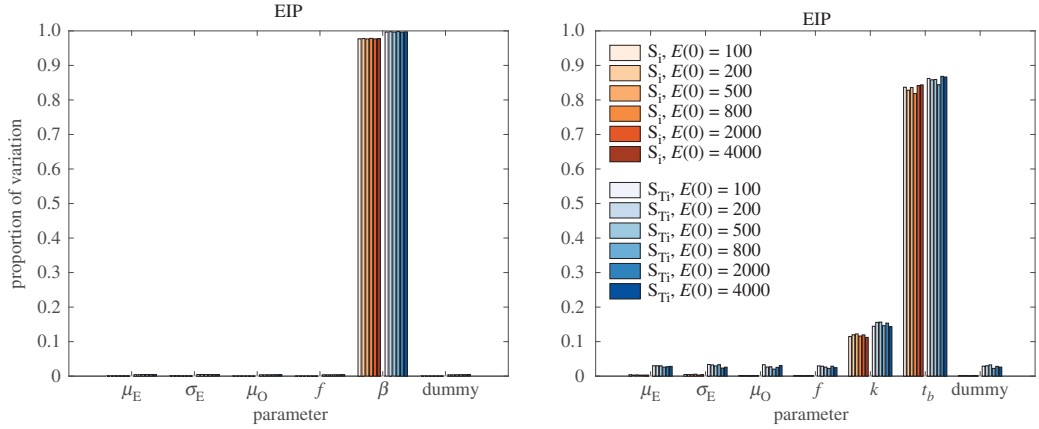

**Figure 7.** Global sensitivity of extrinsic incubation period. Proportion of variation in EIP with respect to each varied parameter in Model 1 (left) and Model 2 (right) using eFAST. Orange bars indicate individual sensitivity index while blue bars are the total sensitivity index. Increasing colour intensity refers to increasing initial ookinete number. The number of oocyst compartments, $N$, is not able to be varied in eFAST. For each initial ookinete density, sensitivity is shown for the optimal $N$: $N = 30$ for $E_0 = 100$, $N = 50$ for $E_0 = 200$, $N = 50$ for $E_0 = 500$, $N = 75$ for $E_0 = 800$, $N = 75$ for $E_0 = 2000$, $N = 75$ for $E_0 = 4000$.

Finally, the variation in $t_b$ and $k$ was clearly correlated. The distribution for $t_b$ and $k$ are highly skewed towards higher values for 500 and 4000 ookinetes. Under both conditions, the 95% HDPI frequently included the upper bound of $k = 10$ (electronic supplementary material, table S4). As discussed above, we expect these values to be positively, but nonlinearly correlated.

## 5.6. Sensitivity of extrinsic incubation period and temporal output

We used extended Fourier amplitude sensitivity test (eFAST) to determine the global sensitivity of EIP, total oocyst count over time, and the sum of each bursting stage over time. This method is able to tell both the proportion of variation by each parameter individually as well as the total sensitivity of each parameter, including all higher order interactions [41,42]. In Model 1, the variation in EIP is dominated by the average time in each bursting oocyst stage, $1/\beta$. This is probably due to the wider range of $\beta$ relative to the variation in this parameter (figure 7, left panel). In contrast, in Model 2, the variation in EIP is dominated by the time to half-maximal bursting, $t_b$, in both individual parameter sensitivity and the total sensitivity index (figure 7, right panel). Interestingly, the rate of maximal bursting, $k$, shows comparatively smaller effect on the variation, approaching that of parameters that do not even directly affect EIP. It is important to note that although only certain parameters directly enter the formula for EIP (equation (4.2)), all parameters are involved in the fitting process. Thus, all parameters can indirectly affect the calculated EIP.

For the temporal output, for Model 1, we find that the ookinete number is controlled mostly by the rates leaving the ookinete compartment, $\mu_E$ and $\sigma_E$ (electronic supplementary material, figure S8). For bursting oocysts and total oocysts, the early time points are dominated by the fraction of bursting oocysts, $f$, and, the rate of transition from ookinetes to oocysts, $\sigma_E$. At later time points, these compartments are impacted by the average time in the bursting oocyst stage, $1/\beta$. In contrast, the non-bursting oocysts are affected by oocyst mortality, $\mu_O$, at later time points. The sporozoite counts are primarily affected by $\beta$. Similar results are seen for the total sensitivity index (electronic supplementary material, figure S9).

For Model 2, the ookinete number is controlled mostly by the rates leaving the ookinete compartment, $\mu_E$ and $\sigma_E$ (electronic supplementary material, figure S10). For oocysts, the rates leaving the ookinete compartment, $\mu_E$ and $\sigma_E$, are important at early time points. At later time points, the timing of half-maximal bursting, $t_b$, followed by the maximal bursting rate, $k$, impact the variation in bursting oocyst count most significantly (electronic supplementary material, figure S10). At later time points, the impact of the variation cannot be truly determined. In contrast, non-bursting oocysts and the total oocysts, are impacted by the oocyst mortality, $\mu_O$ at later time points. The sporozoite count is affected by $t_b$ early on, but by the fraction of bursting oocysts, $f$, and, interestingly, the rate of transition from ookinetes to oocysts, $\sigma_E$, at later time points. Similar results are seen for the total sensitivity index (electronic supplementary material, figure S11).

# 6. Discussion

We build two models of malaria parasite development within the mosquito host and evaluate these models using daily data on oocyst counts and sporozoite measurements from a controlled set of experiments initiated with blood containing standardized ookinete densities [14]. Our Model 1 incorporates a delay to bursting by the inclusion of multiple oocyst stages, while Model 2 includes a time-dependent delay in the oocyst bursting function. With these models, we are able to estimate the extrinsic incubation period, an important metric for determining the probability of parasite transmission from mosquitoes to vertebrate hosts, across varying initial ookinete levels. Because EIP is of the same order of magnitude as the mosquito lifespan, a small decrease in EIP can lead to increased transmission. A key finding of our models is the non-monotonic relationship between initial ookinete numbers and EIP, with intermediate ookinete numbers producing the shortest EIP.

Shorter EIP increases the likelihood that a mosquito will survive the parasite incubation period. While final sporozoite score tends to increase with initial ookinete density, EIP has a non-monotonic relationship with initial ookinete density, with the shortest EIP occurring at intermediate initial ookinete densities. This result suggests the possibility of within-mosquito density-dependent parasite interactions that promote onward transmission to vertebrate hosts at these intermediate initial parasite densities. Without such density-dependent interactions, one might expect EIP to decrease with $E(0)$; however, the lengthening of EIP at higher $E(0)$ indicates that there is probably competition at higher densities that is either prolonging the ookinete stage, delaying oocyst rupture, or increasing oocyst mortality. However, because final sporozoite score increases with $E(0)$ and because there is not an increasing relationship between the proportion of non-bursting oocysts $f$ and $E(0)$, it is unlikely that the density-dependent interactions are resulting in increased oocyst mortality. Likewise, there is no consistent pattern in the duration of the ookinete stage $1/(\sigma_E + \mu_E)$ with increasing $E(0)$. For both Models 1 and 2, however, we do observe a U-shaped pattern in $t^*$, the time following oocyst formation at which the probability of having burst is one-half. Consequently, the effect of density-dependence appears to be on the time between oocyst formation and oocyst bursting.

An alternative hypothesis is that an increase in EIP at high initial ookinete numbers may occur due to a dose-response, such as through increased immune pressure in the presence of many parasites. In this case, we would expect a monotonic increase in the EIP with increasing $E(0)$. This is in contrast to the U-shaped relationship between $E(0)$ and EIP, which we observe and, thus, find density-dependence to be a more parsimonious explanation.

Overall, both Model 1, with an appropriate number of oocyst stages, and Model 2 can fit the oocyst count and sporozoite score data reasonably well. From our results, it is clear that models that lack a delay prior to oocyst bursting will not be able to account for both the timing and output of the sporozoite population. Accuracy in both of these factors is essential for good estimates of transmission. Furthermore, separating bursting from non-bursting oocysts was critical for capturing the continued gradual decrease in oocyst counts beyond the time point at which sporozoite numbers no longer increased. Despite a clear need for separation of bursting and non-bursting oocysts, there remain several parameters that appear to have poor identifiability. In particular, parameters related to the ookinete compartment, $f$, $\mu_E$ and $\sigma_E$, show wide variability. Further, experimental studies that also track early stages of parasite development, such as the formation and loss of the ookinete stage would be needed to more accurately fit these parameters.

While both Model 1 and 2 fit the data quite well, neither Model 1 nor Model 2 definitively described the oocyst bursting process, because each model was selected as the best model for half of the initial ookinete treatments. Thus, our study is unable to confirm whether the time from oocyst formation to oocyst bursting is in fact truly gamma-distributed. However, when Model 1 is selected as the best performing model, Model 2 yields very similar estimates in the parameter $t^*$. This consistency in $t^*$ estimates, but discrepancy in the corresponding estimates of EIP for treatments $E(0) = 100, 200, 2000$, by the two models suggests that the uncertainty in our estimates of EIP arises mainly from uncertainty in the estimation of the ookinete stage duration $1/(\sigma_E + \mu_E)$. Experiments that specifically seek to estimate these parameters through tracking of the ookinete stage could help to reduce the uncertainty in our estimates of EIP.

Despite the fact that Model 1 and Model 2 are each selected as the best model for half of the treatments, the close agreement across several metrics when Model 1 is selected as the best fit model indicates that Model 2 is an appropriate representation of Model 1. Implementing Model 2 in lieu of

Model 1 is computationally advantageous when the differential equation needs to be solved a large number of times (as in the implementation of MCMC), particularly if a large number of bursting oocyst stages $N$ is required.

EIP is sensitive in both Model 1 and Model 2 to parameters directly related to oocyst bursting. In Model 1, this is $\beta$, while in Model 2 this is $t_b$ and to a lesser extent $k$. Such a dominating sensitivity to $\beta$ in Model 1 is probably due to the examination of a significantly larger range within which the value actually varies (we consider 0.5 times the minimum fitted $\beta$ as a lower bound and 2 times the maximum fitted $\beta$ as an upper bound). It is interesting that despite the variation in $\sigma_E$ and $\mu_E$ observed between conditions, the EIP appears highly insensitive to changes in these values. This may be due to the fact that we only perform sensitivity under each initial condition, but not across conditions. The variability across conditions is apparent from our MCMC fits.

# 7. Conclusion

EIP, the time period of pathogen development in the vector until transmissibility, is a key quantity in understanding and evaluating the spread of vector-borne diseases. This quantity remains poorly understood, particularly in diseases such as malaria. A quantitative description of parasite development within the mosquito, as in this work, provides more basis for determining the EIP and which factors most impact its length. Using our quantitative framework, we show that intermediate ookinete densities produce the shortest EIP and the process of bursting produces highest sensitivity to this timing. This knowledge could lead to improved intervention strategies to mitigate disease spread.

Data accessibility. Data and relevant code for this research work are stored in GitHub: https://github.com/laurenchilds/EIP and have been archived within the Zenodo Repository: https://zenodo.org/record/3962371#.Xx8dhfhKhTY.

Authors' contributions. L.M.C. and O.F.P. participated in design of the study, performed data analysis, drafted the manuscript, and gave final approval.

Competing interests. The authors have no competing interests to declare.

Funding. No funding has been received for this article.

Acknowledgements. O.F.P. acknowledges the support of NSF-DMS Award no. 1816075. L.M.C. acknowledges the support of Simons Foundation Collaboration Grant for Mathematicians no. 524390 and NSF-DMS Award no. 1853495. This work was assisted by attendance of L.M.C. as a Short-term Visitor at the National Institute for Mathematical and Biological Synthesis, an Institute supported by the National Science Foundation through NSF Award no. DBI-1300426, with additional support from The University of Tennessee, Knoxville. Any opinions, findings, and conclusions or recommendations expressed in this material are those of the authors and do not necessarily reflect the views of the National Science Foundation. We thank Maria-Gloria Basanez for insightful conversations on the literature and unpublished data. We thank Leah Johnson for helpful conversations on implementation of debInfer.

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
