## [Reviewer comments · Royal Society Open Science]

Review History

RSOS-192173.R0 (Original submission)

Review form: Reviewer 1

Is the manuscript scientifically sound in its present form?

No

Are the interpretations and conclusions justified by the results?

Yes

Is the language acceptable?

Yes

Do you have any ethical concerns with this paper?

No

Have you any concerns about statistical analyses in this paper?

Yes

Recommendation?

Major revision is needed (please make suggestions in comments)

Comments to the Author(s)

This is an interesting manuscript aiming to confront different mathematical models of within-mosquito malaria parasite dynamics with experimental data. The authors succeed in doing this, and gain some interesting insights with relevance for both a better understanding of the biological systems itself, as well as the ways to measure and model this system.

The manuscript would benefit from revisions to (1) clarify methodological details and results, (2) to ensure that all analyses are in fact technically correct, and (3) to tie the quantitative results back to the experimental methods and the biological system itself.

Detailed comments are provided in the attached file (Appendix A).

I am not an expert on multistart optimization, or sensitivity analysis with the eFAST method, so my review focussed on the Bayesian analysis, as well as general aspects of confronting mathematical models with experimental data.

Review form: Reviewer 2

Is the manuscript scientifically sound in its present form?

Yes

Are the interpretations and conclusions justified by the results?

No

Is the language acceptable?

Yes

Do you have any ethical concerns with this paper?

No

Have you any concerns about statistical analyses in this paper?

No

Recommendation?

Major revision is needed (please make suggestions in comments)

Comments to the Author(s)

This work is an important field, and potentially has an interesting contribution to make. As currently presented however, I do have a number of concerns, which are detailed in the attached document (Appendix B).

Decision letter (RSOS-192173.R0)

21-Feb-2020

Dear Dr Prosper,

The editors assigned to your paper ("The impact of within-vector parasite development on the extrinsic incubation period") have now received comments from reviewers. We would like you to revise your paper in accordance with the referee and Associate Editor suggestions which can

be found below (not including confidential reports to the Editor). Please note this decision does not guarantee eventual acceptance.

Please submit a copy of your revised paper before 15-Mar-2020. Please note that the revision deadline will expire at 00.00am on this date. If we do not hear from you within this time then it will be assumed that the paper has been withdrawn. In exceptional circumstances, extensions may be possible if agreed with the Editorial Office in advance. We do not allow multiple rounds of revision so we urge you to make every effort to fully address all of the comments at this stage. If deemed necessary by the Editors, your manuscript will be sent back to one or more of the original reviewers for assessment. If the original reviewers are not available, we may invite new reviewers.

- Data accessibility

If you wish to submit your supporting data or code to Dryad (<http://datadryad.org/>), or modify your current submission to dryad, please use the following link:
<http://datadryad.org/submit?journalID=RSOS&manu=RSOS-192173>

- Competing interests

- Authors' contributions

All submissions, other than those with a single author, must include an Authors' Contributions section which individually lists the specific contribution of each author. The list of Authors should meet all of the following criteria; 1) substantial contributions to conception and design, or

acquisition of data, or analysis and interpretation of data; 2) drafting the article or revising it critically for important intellectual content; and 3) final approval of the version to be published.

- Acknowledgements

- Funding statement

on behalf of Professor Len Thomas (Associate Editor) and Mark Chaplain (Subject Editor)
openscience@royalsociety.org

Associate Editor's comments (Professor Len Thomas):

Thank-you for submitting your manuscript for our consideration. We have now received two reviews, and while both reviewers agree that the manuscript is potentially publishable, both raised major concerns, and so I am recommending you be given the opportunity to revise the manuscript. The first reviewer had concerns about the clarity of your methods description, the methods used and the accuracy of the results. The second had concerns about what is new in this paper vs previous work and about model assumptions. Please, if you revise and resubmit, make sure you address each comment made in the attached reviews, and provide a cover letter detailing how you have dealt with each point. In addition, on your github site, please make sure your data files are clearly documented, in the sense that it is clear what each column of each file represents (perhaps by providing a readme file of some kind). I look forward to seeing your resubmission.

Comments to Author:

Reviewers' Comments to Author:

Reviewer: 1

Comments to the Author(s)

This is an interesting manuscript aiming to confront different mathematical models of within-mosquito malaria parasite dynamics with experimental data. The authors succeed in doing this, and gain some interesting insights with relevance for both a better understanding of the biological systems itself, as well as the ways to measure and model this system.

The manuscript would benefit from revisions to (1) clarify methodological details and results, (2) to ensure that all analyses are in fact technically correct, and (3) to tie the quantitative results back to the experimental methods and the biological system itself.

Detailed comments are provided in the attached file.

I am not an expert on multistart optimization, or sensitivity analysis with the eFAST method, so my review focussed on the Bayesian analysis, as well as general aspects of confronting mathematical models with experimental data.

Reviewer: 2

Comments to the Author(s)

This work is an important field, and potentially has an interesting contribution to make.

As currently presented however, I do have a number of concerns, which are detailed in the attached document.

Author's Response to Decision Letter for (RSOS-192173.R0)

See Appendix C.

Decision letter (RSOS-192173.R1)

Dear Dr Prosper:

On behalf of the Editors, I am pleased to inform you that your Manuscript RSOS-192173.R1 entitled "The impact of within-vector parasite development on the extrinsic incubation period" has been accepted for publication in Royal Society Open Science subject to minor revision in accordance with the referee suggestions. Please find the referees' comments at the end of this email.

The reviewers and Subject Editor have recommended publication, but also suggest some minor revisions to your manuscript. Therefore, I invite you to respond to the comments and revise your manuscript.

- Ethics statement

- Data accessibility

It is a condition of publication that all supporting data are made available either as supplementary information or preferably in a suitable permanent repository. The data accessibility section should state where the article's supporting data can be accessed. This section should also include details, where possible of where to access other relevant research materials

such as statistical tools, protocols, software etc can be accessed. If the data has been deposited in an external repository this section should list the database, accession number and link to the DOI for all data from the article that has been made publicly available. Data sets that have been deposited in an external repository and have a DOI should also be appropriately cited in the manuscript and included in the reference list.

If you wish to submit your supporting data or code to Dryad (<http://datadryad.org/>), or modify your current submission to dryad, please use the following link:
<http://datadryad.org/submit?journalID=RSOS&manu=RSOS-192173.R1>

- **Competing interests**

- **Authors' contributions**

- **Acknowledgements**

- **Funding statement**

Because the schedule for publication is very tight, it is a condition of publication that you submit the revised version of your manuscript before 22-Jul-2020. Please note that the revision deadline will expire at 00.00am on this date. If you do not think you will be able to meet this date please let me know immediately.

When submitting your revised manuscript, you will be able to respond to the comments made by the referees and upload a file "Response to Referees" in "Section 6 - File Upload". You can use this

to document any changes you make to the original manuscript. In order to expedite the processing of the revised manuscript, please be as specific as possible in your response to the referees.

on behalf of Professor Len Thomas (Associate Editor) and Mark Chaplain (Subject Editor)
openscience@royalsociety.org

Associate Editor Comments to Author (Professor Len Thomas):

Associate Editor

Comments to the Author:

Thank-you for your revision and your thorough response to the reviewers' comments. I believe you have accounted well for all comments, and the manuscript is almost ready for acceptance. I have noted a few minor typographical issues, below, which I ask you to address. In addition, I noted your comprehensive response to Reviewer 1 on the matter of using the average values for sporozoite scores, including that you ran the models with both original, arithmetic and "pseudo-geometric" averages -- but I did not see any of this mentioned in the manuscript. I would like to see a small summary of this discussion with the reviewer in the Discussion section, and a more full presentation in Supplemental Materials. Once this is addressed, we should be able to accept the manuscript.

Minor editorial issues:

page 2 line 33. "(See" should be "(see"

page 3 line 24 "being 42 days.Within" Need a space after the period.

page 4 line 10 "The authors consider". Presumably you are referring to the author of [16]? In this case, it should be "The author considers". Okay to use "they" for the rest of the paragraph to avoid assigning gender.

page 7, line 47 "exitflag". I think this should be 2 words "exit flag"

page 7, line 11. AICc as more appropriate for small sample sizes under an assumption of normality (refer to the Burnham and Anderson book); this caveat is worth noting here, perhaps in parentheses.

page 8 line 23. I think "uninformed" should be "uninformative"

page 9, table 2. Please clarify in the caption that the MCMC point estimates shown are posterior medians.

page 11, line 27 - remove comma after k "maximal bursting rage, k, (Table S5"

page 16 figure 2, lower right plot exponent is overwritten by the main title.

Figure 3 and Table S4, you need to be explicit in the captions what level of probability is associated with the HDPIs -- presumably 0.95, so 95% HDPI? Possibly also be explicit about this in the text.

Reviewer comments to Author:

Author's Response to Decision Letter for (RSOS-192173.R1)

See Appendix D.

Decision letter (RSOS-192173.R2)

Dear Dr Prosper,

It is a pleasure to accept your manuscript entitled "The impact of within-vector parasite development on the extrinsic incubation period" in its current form for publication in Royal Society Open Science.

on behalf of Professor Len Thomas (Associate Editor) and Mark Chaplain (Subject Editor)
openscience@royalsociety.org

Associate Editor Comments to Author (Professor Len Thomas):
Associate Editor
Comments to the Author:
Thank-you for making these final changes, as requested.

Appendix A

General Comments: (page numbers refer to those given in the header of the review proof, not those typeset by the journal template)

1. Several parts of the manuscript are rather too terse. This includes the methods section, where details about the experimental and quantitative analyses is incomplete to understand all analyses, but also the final discussion, and many of the figure and table captions. Specific comments are provided below to identify the sections that - in my opinion - would benefit from expansion.
2. I am not entirely convinced that all MCMC analyses ran to convergence. I would strongly suggest that the authors repeat their analyses using multiple MCMC chains with different starting values to ensure convergence was achieved. Similarly - although I am not an expert with multistart methods - I am not convinced that all of these analyses yielded sensible results. The model trajectories plotted in Fig. S3 for high values of the compartment number N shows curves for $E(0)=100,200,500$ and $N=75,100$ that are very poor fits to the data, when most other parameter combinations seem to yield reasonable fits. Is this a parameter identifiability issue for these particular situations, or is this a technical issue with the multistart procedure?
3. In my opinion, the results indicate strong batch effects (i.e. parameter estimates clustering in Fig 6 for the two different experiments, as discussed on page 12 lines 23-27). My suspicion is that there may be an experimentally uncontrolled variable relating to e.g. the quality of the blood, the mosquito cohort, the accuracy with which $E(0)$ was known, or the quality/viability of the ookinetes that has a decided effect on the estimated parasite dynamics. Given this is a re-analysis of a thesis dataset, I appreciate that the authors may not be able to determine the underlying experimental factors for this, but this ought to be at least discussed in the manuscript as it may indicate that the level of replication in the experimental data is insufficient to capture the variability in the system (regardless whether it was caused by experimental/measurement error or natural variation).
4. My understanding is that one of the key innovations of the manuscript is to model different classes of oocytes (bursting/non-bursting). Some parameters relating to the transitions between these classes appear to be only weakly identifiable from the available data (μ_e , σ_e , f). I believe the authors should relate this finding back to the experimental system, and perhaps highlight that more accurate inferences (potentially allowing to get at the unresolved question about the nature of the waiting-time process from oocyst formation to oocyst bursting) might be possible if the two (or more) classes could be experimentally differentiated in future studies.
5. I found the overall organisation of the presentation of results confusing and would suggest consolidating some of the tables and figures, and moving some material from the main text to the supplement and vice versa. I think that in particular it would be helpful to readers with a more experimental background if the raw data (Fig. S1) were presented in the main text. Additional suggestions are given in the specific comments.

Specific comments:

6. Page 3, line 36: I think the fact that the experimental data started infections at the ookinete stage (and not by using blood containing gametocytes) should be restated in the final discussion to highlight where the experimental system may diverge from the natural cycle of the parasite.

7. Page 3, line 47: Please give full bibliographic details for the thesis (University, DOI or a comparable repository url if available). Also note, that a description of the data appears to have been published as Dawes et al. 2009 Transactions Royal Soc Trop Med Hygiene 103:1197-1198

8. Page 4: The data section was very difficult to understand. Given that the data are not formally published it would be helpful if more detail (where available) could be provided here.

8-1: line 10: give the total duration of the experiments (40 days according to Fig S1).

8-2: line 10: give a sample size of mosquitoes dissected each day

8-3: line 12: clarify the term "score". My understanding is this is an "abundance score" is that correct?

8-4: line 13: clarify what the control entailed. Parasite free blood? Is any data from control mosquitoes presented? If not, why not?

8-5: line 19: if the score is related to abundance, some sort of counting must have happened. I'd suggest to change this sentence to "counts of sporozoites were not recorded".

8-6: line 27: clarify "values" do you mean counts/abundances?

8-7: line 31: I think adding Fig S1 to the main text would greatly help in understanding the experiments.

9. Page 4, lines 46-49: Referencing the appropriate panels of Fig S1 would not hurt.

10. Page 4, lines 54-57: This is a key motivation for your study then. State this in the penultimate paragraph of the introduction!

11. Figure 1: I would prefer a more verbose figure caption that explains the symbols for at least some of the different compartments without having to look them up in the text.

12. Table 1: if parameters p and m were not fitted, what value did you use for it in the simulations, and why?

13. Page 6, line 57: Isn't 10 raised to the power of the sporozoite scores? i.e. the scores is the exponent?

14. Page 7, section (b): I believe the parameters that aren't estimated ought to be discussed in this section (see comment #12)

15. Page 8 lines 6-10: I believe Figure S2 is an accidental duplication of another figure (fig 4?) - as I am having trouble to align the text here and in the caption of Fig S2 with the actual figure.)

16. Page 8, section (i). The later results only seem to present AICc values, so why do you state that both AIC and AICc were used? Also, this section is devoid of references when there is a large body of literature about the AIC itself and model selection for dynamical models. Any reasons/justification why you chose this particular approach?

17. Page 9, section (e): Please provide more details on the Bayesian analysis, in particular how MCMC convergence was assessed. Some parameter posteriors appear to either cover a very large proportion of the prior range (e.g. k for $e(0)=500$) and/or are concentrated at the prior bound. Please make sure that multiple chains that have different MCMC starting values do indeed converge to the same posterior. I appreciate that the deBInfer package does not provide the most efficient way of running multiple chains, but the posterior chains for sequential runs of with randomized starting values, can be extracted and joined up using e.g. the coda package to then calculate formal convergence diagnostics as outlined below:

```
#[set up the model and parameters]
#run multiple calls to de_mcmc, ensure that they have the same number of
iterations, and the same parameter set
chain1 <- de_mcmc(...)
chain2 <- de_mcmc(...)
chain3 <- de_mcmc(...)

#combine the samples into an mcmc.list object
all_chains <- coda::as.mcmc.list(list(chain1$samples, chain2$samples,
chain3$samples))
#all_chains can now be manipulated with coda like any multi-chain
analysis, e.g.
plot(all_chains)
gelman.diag(all_chains)

#To harness the samples from all chains in the deBInfer plotting
functions and posterior trajectory simulations etc. one can manipulate an
existing debinfer_result object:
#NB: the chains will be merged using this approach

merged_results <- chain1 #make a copy of the output structure
merged_results$samples <- coda::as.mcmc(do.call(rbind, all_chains))

post_prior_densplot(merged_results, burnin = 1)
```

18. Table 2: I would consolidate this table into table 1 for conciseness.

19. Table 3: I think it would be useful to add the credible intervals for each parameter for the Bayesian results. (And also use them in the subsequent discussion of how similar multistart and Bayesian results are, i.e. do the former fall within some posterior credible interval of the latter). I would also present the relevant AICc values in this table.

20. page 11. line 7: Give explicit AICc differences here. The y-axis scale in Figure 3 is so large (and variable among the different fits)

that it is hard to see how similar the two different models performed exactly

21. When discussing similarity of parameter estimates make use of the uncertainty estimates from the Bayesian fit (cf comment#19)

22. Page 4, line 18: Expand what you mean by "reasonable fit". Some of the fits in Fig S3, do indeed look not reasonable (see comment #2). Is this an issue of concern?

23. Page 11, line 31: Figure 7 is a bit crowded and the ordering of peaks is not easy to see. Can you add panels that plot the estimated time of the peak oocyst count / the estimated final sporozoite score against $E(0)$?

24. The "wrong" ordering of the peaks may well be an experimental artefact caused by a batch effect of some sort (see comment #3). This should be expanded on here and/or in the discussion, especially as - as far as I understand - the dependence structure of the different experiments was not explicitly modelled (and probably can't be given the lack of replication).

25. Page 11, lines 34-36: I am not clear about the point you are trying to make in the sentence starting "Thus, interestingly". Can you please clarify this sentence, and the link between f and the other estimated quantities?

26. page 12, first paragraph: I think there may be a parameter identifiability issue here that deserves mention, which is that you are trying to estimate transitions between latent classes of which you can only observe a sum count. Does this have implications for future experiments? (I think so. See comment #4)

27. Line 17/Figure 6. Please ensure that the variation you describe is not an artefact of an uncovered MCMC!

28. Page 12, lines 23-27: Again, I believe that batch effects in the experiment are at work here.

29. Page 13, Line 22: This is the place, where you should - in my opinion - tie the models back to the biological system and it's experimental treatment, and mention that further developments of the experimental approach are needed to check whether these classes of oocysts are indeed distinguishable in the real world.

30. Page 13, line 30: I am not an expert on mosquito-plasmodium interactions, but besides density dependent competition among the parasites couldn't there be also a dose-response relationship at play where the mosquito host reacts differently to different parasite concentrations?

32. Page 13, Line 47: Again, wouldn't it be nice if there was experimental data to allow a better separation of the estimates of μ_E and σ_E ? Do state this. Confronting models with data is a two-way

street, and you should provide take-home messages for future experimentators on how they might ensure the collection of data that could better inform the models that you present.

33. Page 14, Section 7: See comment #32. I think you should put some take-home messages for future experiments here. Justify once more why you think understanding the waiting-time process matters biologically/epidemiologically and how that is related to your assumption of different oocyte classes, and then issue a call for experiments to identify these classes.

34. Fig. 2: I would suggest moving this to the supplement.

35. Fig 5: Can you add posterior credible intervals around the presented MCMC draws?

36. Fig. 7: see comments #23

37. Fig S1: I would suggest moving this into the main text.

38. Fig S2: I think the figure shown here is not the correct figure, but a duplicate of Fig 4.

Appendix B

Reviewer's Notes Re "The Impact of within-vector parasite development on the extrinsic incubation period."

Note, page numbers are as per pdf, top left of page.

This paper seeks to progress knowledge regarding the EIP of Plasmodium in Anopheles vector mosquitoes. This is an important area, with much scope for improved understanding. The authors have clearly done a lot of work, and have some novel contributions to report.

I have a number of concerns, however, about the manuscript in its present form.

1. The key results from the modelling exercise need to be stated more clearly, in terms of the biological insights they provide.
2. More transparency is needed regarding the advances made here compared to the original work on which this exercise is based.

Detailed points are listed below, in no significant order.

1. Source literature

The work presented in the *ms* is built on data scraped from a 2011 doctoral thesis by Emma Dawes. The *ms* cites this thesis with the comment 'To examine this, a series of experiments measuring oocyst and sporozoite density of Plasmodium berghei in Anopheles stephensi mosquitoes were performed but never formally published [12]'.

However, as explained in the thesis (See, for example p108 and Appendix C), experiments and models contributing to the thesis were formally published and analysed in a number of peer-reviewed papers. The relevant papers should be cited here, including, for example, [1-5]. In so far as these include models, they should also be added to the list (page 3 line 42) of references re. mathematical work on parasite dynamics in the mosquito. It should also be made clear that this list is illustrative not exhaustive.

2. The sporozoite score metric

Rows 19-20 page 4 "sporozoite levels were recorded by a score between 0 and 4 on an exponential scale." Firstly, this is, if anything, a log scale, not an exponential scale. Similarly, page 6, line 58, "*si* is the data of sporozoite count (taken by raising the sporozoite score to the power ten)", is incorrect, this should presumably be ten raised to the power of the sporozoite score?

Associated with this query, I have concerns around the use of averages of the discrete scoring values. From Fig 6.3 in the thesis, it is clear that the scores are allocated as integer values, each representing the ranges repeated in this *ms*, ie, a score of 0 means no sporozoites in the salivary glands, a score of 1 means 1 to 10, of 2 means 11 to 100, 3 means 101 to 1000, 4 means 1001 to 10000 and so on.

Taking the actual scores as powers of 10 gives the maximum, not the average, sporozoite count for each score.

The current *ms* uses average values for the scores, scraped from a figure in the thesis. It is important to understand how the original averages were calculated in order to understand what these values actually represent, whether they are correctly interpreted and used in the current modelling exercise, and how the necessity of using data based on the scoring scale may affect the model results. Note there is some information on page 182 and in table 6.2 of the thesis, but to add to the confusion, the figures labelled in this table as geometric means are not in fact geometric means, but rather 10 raised to the power of the arithmetic means of successive scores.

We need a brief but clear explanation of how/whether this unusual metric does/does not impact the results and/or conclusions in this *ms*.

3. Relationship between parameters f and μ_d

The authors have chosen to assume that the ‘bursting’ category of oocysts includes only those oocysts which actually survive to burst, and so no mortality is applied to this category during the period between oocyst formation and oocyst bursting. The number of oocysts placed in this category is calculated as a fixed proportion of the oocysts generated from ookinetes, parameter ‘ f ’. This could reflect an assumption that oocysts capable of bursting do not experience mortality, in which case the proportion of newly formed oocysts falling into this category is independent of the mortality subsequently experienced by oocysts in the non-bursting category. Alternatively, all oocysts may be subject to mortality, with f representing the number of oocysts which ultimately survive to burst, as a proportion of the initial oocysts generated from ookinetes. In the latter case both ‘ f ’ and the mortality rate for non-bursting oocysts, μ_d , would be functions of the overall oocyst mortality rate. In the model, f and μ_d are treated as independent fixed values.

The authors need to explain and justify the assumptions underlying their choices for this aspect of the model regarding the underlying biology and/or any impact on the results or the interpretation of the results. They also need to reflect this properly in their commentary on correlation between model variables, replacing the current rather confusing text, from p11 line 58, ‘We also observed strong, negative correlations between f and μ_o . As only non-bursting oocysts are subject to oocyst mortality, μ_o , when f is higher fewer oocysts enter the O_d class, so to persist long enough they must be subjected to lower mortality.’

4. Novelty of non-bursting oocyst category concept

The authors mention the concept of a non-bursting category for the oocysts as a key novel component in their analysis, eg. “The idea of two oocyst sub-populations, one which bursts and one which does not, has not, to our knowledge, been discussed before.” Note that Fig 6.1 in the thesis, which presents a flowchart of *Plasmodium* progression through the vector, has a non-bursting oocyst category ‘remain on midgut’, and there is discussion in the thesis text relating to this (eg pages 162, 177), although I think the *ms* is correct in saying that this concept is not reflected in the mathematical models developed in the thesis.

5. Clarity of Table 3

Page 11 lines 11-12 “A summary of parameter estimates is provided in Table 3; the best overall model as determined by multistart is indicated in bold.” In my copy, the emboldening in Table 3 is not visible.

6. Previous models in thesis and associated papers vs models in this *ms*

Page 9 lines 55 & 56, “Previous efforts have fit the oocyst data and the sporozoite data independently [12]. However, none of the proposed models are able to fit both data sets simultaneously (not shown).” also page 4 lines 51 to 57, “Our modeling is motivated by the work of [12]. The authors consider two successive model structures: (i) multiple ookinete stages to produce a single oocyst stage and (ii) multiple oocyst stages starting at day 10 to produce sporozoites. They fit (i) to oocyst data and fit (ii) to sporozoite data. Although we find excellent replicability of their results assuming a similar model structure (not shown), we find that neither of their models alone can fit both the oocyst and sporozoite data simultaneously. Here, we build two models to account for both oocyst and sporozoite data together.”

In both cases the description in the *ms* of the thesis models might readily be misinterpreted to mean that the models referred to attempted to fit both data sets, but could only correctly fit one or the other, indicating some inconsistency within the models. In fact, as stated, the thesis models each model only one of the two transitions, and it is therefore meaningless to say that they fail to fit both. It would probably be sufficient here for the authors to indicate that previous models were designed to explore the relevant transitions separately, whilst they now model both transitions within a single framework.

Having created a model which does include both transitions, the authors subsequently say (page 13 from line 39) “Neither Model 1 nor Model 2 definitively described the oocyst bursting process, because each model was selected as the best model for half of the initial ookinete treatments. Thus, our study is unable to confirm whether the time from oocyst formation to oocyst bursting is in fact truly Gamma-distributed. However, when Model 1 is selected as the best performing model, Model 2 yields very similar estimates in the parameter t^* . This consistency in t^* estimates, but discrepancy in the corresponding estimates of EIP for treatments $E(0) = 100, 200, 2000$, by the two models suggests that the uncertainty in our estimates of EIP arises mainly from uncertainty in the estimation of the ookinete stage duration $1/(E + \mu E)$. Experiments that specifically seek to estimate these parameters could help to reduce the uncertainty in our estimates of EIP.” I am not clear exactly what is meant here. Does it mean that these models also fail to fit both transitions simultaneously?

In general, it would be helpful for the authors to be more explicit about the extent to which their models build from the models in the original work. For example, the thesis uses a compartmental model with a gamma distribution for the oocyst to salivary gland sporozoite transition, so Model 1 in the current *ms* equates to the thesis model with the key addition of the non-bursting category.

7. Repeated use of parameter label m

The models have a fairly small number of parameters, it seems unhelpful therefore to use ' m ' to represent two different variables (number of sporozoites emerging from oocyst in models 1 and 2, and total number of time points in the maximum likelihood calculation)

8. Use of logistic function for distribution of EIP

The time-dependent function used in Model 2 for the rate at which oocysts burst is a logistic function. This is consistent with previous studies which fitted logistic functions to the distribution of EIP timings for *Plasmodium* development in *Anopheles*, and the previous work should be referenced here, eg [6-8]

9. Figure 2

Is Figure 2 referenced in the text?

What is this figure intended to convey, why are curves shown for multiple parameter values of model 1, but a single parameter set for Model 2???

10. Context of results in past work

Page 11 from line 29 '(b) Oocyst count and sporozoite score with increasing ookinete density'. The authors need to compare their results to those in [1] and in the thesis including eg. Figures 6.14, 6.20, 6.21

11. Section headed 'EIP variation and identifiability'

From page 12 line 53 'EIP variation and identifiability'

Firstly, this section discusses correlations between various model parameters and output, and is not on the whole well-described by the section title.

As a general point the correlations discussed need to be better described in terms of the insights they offer regarding the biology being modelled, or else what they indicate of interest regarding any limitations of the model. At present the text here appears confused and is not telling a clear story about the meaning of the observed correlations.

See separate comment re relationship between f and μ_0 .

12. Figure 8.

(i) Why is the x axis for Figure 8 labelled 'time, days' when the axis labels are categorical?

(ii) Why is the 'N' parameter not included in the list of sensitised parameter values?

13. Defined relationships between EIP and model parameters

Page 12 from line 38, and Page 13, from line 54. The authors define EIP as equal to the average time in the ookinete category plus the time taken for 50% of bursting oocysts to have burst. Stating that the parameters which primarily explain variation in the EIP are the parameters which specifically determine these two time periods is therefore surely merely stating the self-evident, isn't this an inevitable outcome of the definitions rather than an informative model result? If it does need to be stated, it definitely does not seem worth stating twice.

14. Conclusion

Page 14, the section headed ‘Conclusion’ seems to state generalities, EIP is important and poorly understood, better understanding of EIP could support improved disease control. This states the context for the work presented in the *ms*, but offers no information about the work and results presented here.

The conclusion needs to provide a succinct summary of what this modelling exercise has achieved.

1. Sinden RE, Dawes EJ, Alavi Y, Waldock J, Finney O, Mendoza J, Butcher GA, Andrews L, Hill AV, Gilbert SC: **Progression of *Plasmodium berghei* through *Anopheles stephensi* is density-dependent.** *PLoS Pathogens* 2007, **3**:2005 - 2016.
2. Churcher TS, Dawes EJ, Sinden RE, Christophides GK, Koella JC, Basáñez M-G: **Population biology of malaria within the mosquito: density-dependent processes and potential implications for transmission-blocking interventions.** *Malaria Journal* 2010, **9**:311.
3. Dawes EJ, Zhuang S, Sinden RE, Basáñez M-G: **The temporal dynamics of *Plasmodium* density through the sporogonic cycle within *Anopheles* mosquitoes.** *Transactions of the Royal Society of Tropical Medicine and Hygiene* 2009, **103**:1197-1198.
4. Dawes E, Churcher T, Zhuang S, Sinden R, Basanez M-G: ***Anopheles* mortality is both age- and *Plasmodium*-density dependent: implications for malaria transmission.** *Malaria Journal* 2009, **8**:228.
5. Basáñez M-G FJ, Dawes EJ, Finney O, Sinden RE: **The impact of parasite density on life-stage transitions of *Plasmodium* within the mosquito vector.** *Proceedings of the 11th International Congress of Parasitology* 2006:11-18.
6. Paaijmans KP, Blanford S, Chan BHK, Thomas MB: **Warmer temperatures reduce the vectorial capacity of malaria mosquitoes.** *Biology Letters* 2011.
7. Shapiro LL, Murdock CC, Jacobs GR, Thomas RJ, Thomas MB: **Larval food quantity affects the capacity of adult mosquitoes to transmit human malaria.** *Proceedings of the Royal Society B: Biological Sciences* 2016, **283**:20160298.
8. Shapiro LLM, Whitehead SA, Thomas MB: **Quantifying the effects of temperature on mosquito and parasite traits that determine the transmission potential of human malaria.** *PLoS biology* 2017, **15**:e2003489.

Appendix C

We thank both reviewers for their extremely valuable and detailed feedback. Major changes to the manuscript include:

1. Extension of our MCMC procedure to include five chains, each of at least 100,000 steps for the six different initial ookinete number. This required the modification of all MCMC related figures (see #2).
2. Modifications of many of the figures and captions, as requested by the reviewers. Details of these changes are described in detail below. In addition, the AICc plot for $E(0) = 4000$ has been corrected (an old version was mistakenly included in the original submission). The shape of the curve is qualitatively the same as before, but the AICc values are larger. AICc values are now also included in a table in the supplement. The eFAST plots have been corrected (old versions were mistakenly included in the original submission). The plots are qualitatively the same as before and quantitatively nearly indistinguishable from those in the original submission.
3. Enhanced discussion of the methodology included convergence diagnostics for both multistart and MCMC. We have included two additional tables in the supplement including these convergence metrics. We also note that all fitting results are available in our GitHub Repository where additional information on convergence can be examined.
4. Inclusion of highest density posterior intervals for all parameters, found in a table in the supplement. In addition, some parameter values in Table 2 have been corrected (incorrect values for several parameters from multistart Model 1 were mistakenly included in the original manuscript).
5. Revised and more in-depth discussion of the experimental methodology and previous work performed by Dawes and co-authors. This also includes references to the relevant publications.
6. Improved description and discussion of results pertaining to parameter correlations and variability. This includes the addition of a supplemental table of the 95% highest density posterior intervals for parameters from the MCMC chains. We also note that the chains are available in our GitHub Repository.
7. The code on our GitHub Repository has been cleaned and documented in a ReadMe file.
8. Correction of all minor errors noted by the reviewers.

Please find our responses to the reviewers' specific comments below. We have replicated both reviews here and address points individually in-line. Our responses are red. Additionally, all changes in the manuscript are highlighted in red.

Reviewer's Notes Re "The Impact of within-vector parasite development on the extrinsic incubation period."

Note, page numbers are as per pdf, top left of page.

This paper seeks to progress knowledge regarding the EIP of Plasmodium in Anopheles vector mosquitoes. This is an important area, with much scope for improved understanding. The authors have clearly done a lot of work, and have some novel contributions to report.

I have a number of concerns, however, about the manuscript in its present form.

1. The key results from the modelling exercise need to be stated more clearly, in terms of the biological insights they provide.
2. More transparency is needed regarding the advances made here compared to the original work on which this exercise is based.

Detailed points are listed below, in no significant order.

1. Source literature

The work presented in the *ms* is built on data scraped from a 2011 doctoral thesis by Emma Dawes. The *ms* cites this thesis with the comment 'To examine this, a series of experiments measuring oocyst and sporozoite density of Plasmodium berghei in Anopheles stephensi mosquitoes were performed but never formally published [12]'

However, as explained in the thesis (See, for example p108 and Appendix C), experiments and models contributing to the thesis were formally published and analysed in a number of peer-reviewed papers. The relevant papers should be cited here, including, for example, [1-5]. In so far as these include models, they should also be added to the list (page 3 line 42) of references re. mathematical work on parasite dynamics in the mosquito. It should also be made clear that this list is illustrative not exhaustive.

We thank the reviewer for this note. We have clarified in our manuscript that much of the experimental data presented in the Dawes thesis has been published, including a summary figure of some temporal oocyst data for the experiments conducted in Ch. 4 of the thesis (reference [3] below), and have cited these references accordingly. However, we were unable to find a published manuscript presenting the oocyst nor the sporozoite data that we extracted from Ch. 6 of the thesis for our model fitting. We note that the mean oocyst data published in ref [3] does not coincide with the mean oocyst data we extracted from Figure 6.14 on p. 181 of the thesis; our understanding from the thesis is that the published figure is based on preliminary oocyst count data, whereas the data we extracted from the thesis encompasses the full experimental data.

We agree that the published models emerging from the thesis should be cited along with those already cited in our manuscript; to this end, we have added references [2] and [3] listed below as examples of models of within-mosquito Plasmodium dynamics.

2. The sporozoite score metric

Rows 19-20 page 4 “sporozoite levels were recorded by a score between 0 and 4 on an exponential scale.” Firstly, this is, if anything, a log scale, not an exponential scale.

We have changed “exponential” to “log” scale.

Similarly, page 6, line 58, “*si* is the data of sporozoite count (taken by raising the sporozoite score to the power ten)”, is incorrect, this should presumably be ten raised to the power of the sporozoite score?

The reviewer is correct – our phrasing is reversed; we have made the correction accordingly.

Associated with this query, I have concerns around the use of averages of the discrete scoring values. From Fig 6.3 in the thesis, it is clear that the scores are allocated as integer values, each representing the ranges repeated in this *ms*, ie, a score of 0 means no sporozoites in the salivary glands, a score of 1 means 1 to 10, of 2 means 11 to 100, 3 means 101 to 1000, 4 means 1001 to 10000 and so on.

Taking the actual scores as powers of 10 gives the maximum, not the average, sporozoite count for each score.

The current *ms* uses average values for the scores, scraped from a figure in the thesis. It is important to understand how the original averages were calculated in order to understand what these values actually represent, whether they are correctly interpreted and used in the current modelling exercise, and how the necessity of using data based on the scoring scale may affect the model results. Note there is some information on page 182 and in table 6.2 of the thesis, but to add to the confusion, the figures labelled in this table as geometric means are not in fact geometric means, but rather 10 raised to the power of the arithmetic means of successive scores. We need a brief but clear explanation of how/whether this unusual metric does/does not impact the results and/or conclusions in this *ms*.

Using the reported score as powers of 10 does give a maximum of sporozoites present. We also considered the two other ways to scale from the sporozoite score listed in Table 2 of the Dawes thesis: (i) arithmetic and (ii) “pseudo”-geometric mean. As the reviewer notes, it is not truly a geometric mean; hence, we refer to this version as “pseudo-geometric.” As the average scores are not whole number, we use these methods as follows. For (i) arithmetic, we take the arithmetic mean from $10^{\text{floor}(\text{average score})}$ to $\text{round}(10^{\text{average score}})$. Thus, if the average score was 1.8, then we would consider the arithmetic mean of the numbers from 10 to 63 or a value of 36.5. For (ii) pseudo-geometric, we raised 10 to the arithmetic mean of the score and the floor(score). Thus, if the average score was 1.8, then we would consider 10 raised to the average value of 1 and 1.8 or $10^{1.4}$, a value of 25.1. Using these procedures, we ran a 10 set multistart

(i.e. fitting with 10 different starting parameter sets) for each of $N = 2, 3, 10, 20, 30, 40, 50, 75,$ and 100 for Model 1 as well as a 10 set multistart run for Model 2. These are the same choices for multistart as in the original manuscript. Our results were qualitatively quite similar under the three scenarios. We produce the EIP plot (one of our key results) here with the three versions.

As the average score falls between whole number scores, it's impossible to know if the arithmetic and pseudo-geometric mean calculations are over or under-estimating the true values. While for the 10 raised to the average score will necessarily overestimate the value, we choose the latter.

3. Relationship between parameters f and μd

The authors have chosen to assume that the 'bursting' category of oocysts includes only those oocysts which actually survive to burst, and so no mortality is applied to this category during the period between oocyst formation and oocyst bursting. The number of oocysts placed in this category is calculated as a fixed proportion of the oocysts generated from ookinetes, parameter ' f '. This could reflect an assumption that oocysts capable of bursting do not experience mortality, in which case the proportion of newly formed oocysts falling into this category is independent of the mortality subsequently experienced by oocysts in the non-bursting category. Alternatively, all oocysts may be subject to mortality, with f representing the number of oocysts which ultimately survive to burst, as a proportion of the initial oocysts generated from ookinetes. In the latter case both ' f ' and the mortality rate for non-bursting oocysts, μd , would be functions of the overall oocyst mortality rate. In the model, f and μd

are treated as independent fixed values.

The authors need to explain and justify the assumptions underlying their choices for this aspect of the model regarding the underlying biology and/or any impact on the results or the interpretation of the results. They also need to reflect this properly in their commentary on correlation between model variables, replacing the current rather confusing text, from p11 line 58, 'We also observed strong, negative correlations between f and μO . As only non-bursting

oocysts are subject to oocyst mortality, μO , when f is higher fewer oocysts enter the Od class, so to persist long enough they must be subjected to lower mortality.’

In general, oocyst mortality is considered very low. In fact, most models, which focus on *Plasmodium falciparum* rather than *P. berghei*, ignore oocyst mortality completely (e.g. Teboh-Ewungkem & Yuster, Journal of Theoretical Biology, 2010; Teboh-Ewungkem et al, In: Infectious Disease Modelling Research Progress, 2009; Childs & Prosper, PLOS One, 2017). However, it was clear from the data that oocyst loss was occurring as there were notable decreases in the oocyst numbers while the sporozoite score remained unchanged (or decreased).

We consider f to be the proportion of transitioning oocysts that become bursting oocysts and μ_O to be the mortality rate of non-bursting oocysts. We have chosen these definitions to avoid the complication of rates as mentioned by the reviewer. We do not model the mortality rate of oocysts explicitly. Thus, we consider these to be independent and fit them separately.

We do observe correlations between f and μ_O , as discussed in the revised and renamed “Parameter Correlations” section. Mathematically, this correlation is expected. The level of non-bursting oocysts is entirely dictated by the rate entering $((1-f)*\sigma_E)*E(t)$ and the rate exiting $\mu_O*Od(t)$. Thus, higher f (i.e. lower $1-f$) necessitates a lower μ_O for some oocysts to persist the entire length of the experiments, a maximum of 42 days in some cases.

4. Novelty of non-bursting oocyst category concept

The authors mention the concept of a non-bursting category for the oocysts as a key novel component in their analysis, eg. “The idea of two oocyst sub-populations, one which bursts and one which does not, has not, to our knowledge, been discussed before.” Note that Fig 6.1 in the thesis, which presents a flowchart of *Plasmodium* progression through the vector, has a non-bursting oocyst category ‘remain on midgut’, and there is discussion in the thesis text relating to this (eg pages 162, 177), although I think the *ms* is correct in saying that this concept is not reflected in the mathematical models developed in the thesis.

To address the reviewer’s concern, and to clarify our contribution, we have changed the word “discussed” to modeled in the sentence, “*The idea of two oocyst sub-populations, one which bursts and one which does not, has not, to our knowledge, been modeled before.*”, and we have added a brief note acknowledging that the thesis made a similar observation that not all oocysts will either rupture or die by the end of the experiment.

The flowchart in Fig 6.1 of the thesis does indicate different possible fates of oocysts (rupturing, dying, and remaining on the midgut); however, it does not explicitly suggest separating oocysts into the categories of bursting and non-bursting; i.e. there is no separate compartment emerging from the ookinete compartment indicated for non-bursting oocysts. Instead, the flowchart motivates a more typical model construction in which a single oocyst class has multiple possible fates.

It is worth mentioning that we considered several model formulations before arriving at the two presented in this manuscript, one of which has a single oocyst compartment subject to both

bursting and mortality; we determined that such a construction is incapable of simultaneously fitting to the oocyst and sporozoite data, as well as capturing the observation made in the thesis on p. 162, that many oocysts will remain on the midgut by the end of the experiment. This is where the idea for separating bursting from non-bursting oocysts in our modeling framework arose, and this is what we claim to be a novel contribution to the body of work on modeling within-mosquito *Plasmodium* dynamics.

5. Clarity of Table 3

Page 11 lines 11-12 “A summary of parameter estimates is provided in Table 3; the best overall model as determined by multistart is indicated in bold.” In my copy, the emboldening in Table 3 is not visible.

We have replaced the emboldening with a star.

6. Previous models in thesis and associated papers vs models in this *ms*

Page 9 lines 55 & 56, “Previous efforts have fit the oocyst data and the sporozoite data independently [12]. However, none of the proposed models are able to fit both data sets simultaneously (not shown).” also page 4 lines 51 to 57, “Our modeling is motivated by the work of [12]. The authors consider two successive model structures: (i) multiple ookinete stages to produce a single oocyst stage and (ii) multiple oocyst stages starting at day 10 to produce sporozoites. They fit (i) to oocyst data and fit (ii) to sporozoite data. Although we find excellent replicability of their results assuming a similar model structure (not shown), we find that neither of their models alone can fit both the oocyst and sporozoite data simultaneously. Here, we build two models to account for both oocyst and sporozoite data together.”

In both cases the description in the *ms* of the thesis models might readily be misinterpreted to mean that the models referred to attempted to fit both data sets, but could only correctly fit one or the other, indicating some inconsistency within the models. In fact, as stated, the thesis models each model only one of the two transitions, and it is therefore meaningless to say that they fail to fit both. It would probably be sufficient here for the authors to indicate that previous models were designed to explore the relevant transitions separately, whilst they now model both transitions within a single framework.

We have clarified our language to make it clear that the models presented in the thesis were designed with distinct goals in mind, that differ from those in our manuscript. In particular, we mention that one model from the thesis models the ookinete-to-oocyst transition, which inherently can only be fit to the oocyst data because there is no sporozoite compartment. The second model from the thesis is an oocyst-to-sporozoite model, which can, in principle, be fit to both oocyst and sporozoite data simultaneously, but this was not the goal of the thesis. We conducted this latter experiment to determine whether the thesis’ oocyst-to-sporozoite model could be used for our purposes of studying EIP, and discovered that it was unable to capture the salient features of both the oocyst and sporozoite dataset.

Having created a model which does include both transitions, the authors subsequently say (page 13 from line 39) “Neither Model 1 nor Model 2 definitively described the oocyst bursting process, because each model was selected as the best model for half of the initial ookinete treatments. Thus, our study is unable to confirm whether the time from oocyst formation to oocyst bursting is in fact truly Gamma-distributed. However, when Model 1 is selected as the best performing model, Model 2 yields very similar estimates in the parameter t^* . This consistency in t^* estimates, but discrepancy in the corresponding estimates of EIP for treatments $E(0) = 100, 200, 2000$, by the two models suggests that the uncertainty in our estimates of EIP arises mainly from uncertainty in the estimation of the ookinete stage duration $1/(\lambda E + \mu E)$. Experiments that specifically seek to estimate these parameters could help to reduce the uncertainty in our estimates of EIP.” I am not clear exactly what is meant here. Does it mean that these models also fail to fit both transitions simultaneously?

We thank the reviewer for this question. In short, no – *both* of our models are able to provide very good fits to the oocyst and sporozoite data simultaneously. What we intended to convey is that because both models provide very good, and often very similar, fits to the combined data, with Model 1 outperforming Model 2 for exactly half of the experiments, we are unable to determine which is the *better* model *overall* – one with gamma-distributed time to oocyst bursting (model 1), or one with a time-dependent bursting function (model 2). We make the argument that ookinete data may be able to help us to determine whether one model is better than the other. We have revised the text to help clarify.

In general, it would be helpful for the authors to be more explicit about the extent to which their models build from the models in the original work. For example, the thesis uses a compartmental model with a gamma distribution for the oocyst to salivary gland sporozoite transition, so Model 1 in the current ms equates to the thesis model with the key addition of the non-bursting category.

Our Model 1 includes ookinetes, bursting oocysts with gamma-distributed rupturing, non-bursting oocysts (with exponentially distributed mortality), and sporozoites, whereas the model in Ch. 6 of the thesis includes a gamma-distributed oocyst stage and a sporozoite stage. The two primary models of Ch. 6 in the thesis were described in the second paragraph of section 3 in our manuscript, but we have now emphasized that the multiple ookinete compartments of the first model, and multiple oocyst compartments of the second model, lead to gamma-distributed waiting times.

The details of our two models can be found in sections 3(a) and 3(b) of the manuscript.

It is well-known that a gamma-distributed sojourn time in a compartmental model can be captured by dividing the desired compartment into a series of stages with equal, exponentially distributed duration. We have added some references related to this topic. We knew that we needed a means to create the delay between oocyst formation and oocyst bursting, and that an exponentially distributed rupturing oocyst stage would be unable to capture that delay. One way to accomplish this delay is with a time-dependent rupture function (Model 2), and another is to incorporate the gamma-distributed time to bursting in the oocyst stage (Model 1; see also Figure

S2 illustrating the effect of changing the scale parameter, analogous to changing the number of oocyst stages, on the PDF and CDF of the gamma distribution).

7. Repeated use of parameter label m

The models have a fairly small number of parameters, it seems unhelpful therefore to use ‘ m ’ to represent two different variables (number of sporozoites emerging from oocyst in models 1 and 2, and total number of time points in the maximum likelihood calculation)

We thank the reviewer for catching this oversight. We have changed the ‘ m ’ in the maximum likelihood calculation to ‘ M_O ’ and ‘ M_S ’ to represent the number of time points in the oocyst and sporozoite data, respectively.

8. Use of logistic function for distribution of EIP

The time-dependent function used in Model 2 for the rate at which oocysts burst is a logistic function. This is consistent with previous studies which fitted logistic functions to the distribution of EIP timings for *Plasmodium* development in *Anopheles*, and the previous work should be referenced here, eg [6-8]

The reviewer is correct that logistic functions have been fitted to data related to EIP in previous work. However, it is important to note that while the functional forms are the same, the contexts and comparisons to data are distinctly different from the use of the logistic function in our manuscript. As the reviewer noted, our logistic function represents bursting rate as a function of time – a within-mosquito process. On the other hand, in references [6-8] below, the logistic function represents the cumulative change in proportion of infectious mosquitoes from a group of mosquitoes. Thus, while this (very broadly used) functional form is the same, they represent distinctly different processes. Moreover, we do not fit our bursting rate function directly to bursting rate data; we instead fit the full differential equation parasite life-cycle model (Model 2), to oocyst and sporozoite data. References [6-8], on the other hand, fit the logistic function directly to data on the cumulative change in proportion of infectious mosquitoes, a mosquito population-level metric, rather than a within-mosquito *Plasmodium* parasite population-level metric. In other words, the interpretation of the parameters estimated in our sigmoid function, versus those in the sigmoid functions of [6-8] are fundamentally different. We also want to emphasize that references [7-8] use sporozoite prevalence data (i.e. proportion of population of mosquitoes with sporozoites present in salivary glands), not sporozoite abundance data (i.e. quantity of sporozoites within a single mosquito) as we do in our manuscript.

From our parameter estimates resulting from fitting of the differential equation models to parasite data, we can then calculate EIP, but it does not result directly from fitting a logistic function to some data.

We do feel that references [6-8] are worth including in our introduction to point readers towards other approaches in the literature related to estimating EIP.

9. Figure 2

Is Figure 2 referenced in the text?

What is this figure intended to convey, why are curves shown for multiple parameter values of model 1, but a single parameter set for Model 2???

Former Figure 2 (now Figure S2 in the supplement as per the suggestion of Reviewer 2) shows the quantitative differences resulting from the structure in Model 1 and Model 2. Although the multiple oocyst compartments (as N increases) appears to approximate the CDF more appropriately, there remain differences in the PDF. We show Model 1 for different N values, i.e. different number of bursting oocyst stages. This is one method for mimicking a time delay, as described above in our response to #6. For Model 2, we instead use a time dependent bursting function. There is no N parameter in Model 2. We have added to the Model Formulation and Selection Section to discuss this, which includes a reference to Figure S2.

10. Context of results in past work

Page 11 from line 29 ‘(b) Oocyst count and sporozoite score with increasing ookinete density’. The authors need to compare their results to those in [1] and in the thesis including eg. Figures 6.14, 6.20, 6.21

We have added to this section to address comparisons to 6.14 and 6.21. It is important to note that we are fitting the entire set of ookinete to sporozoite transitions, while their goal in these figures is only to examine the ookinete to oocyst transition. Dawes (2011) observed an increase in peak height and a shift in peak location to the right in Figure 6.21 when considering what appears to be a fixed parameter set and varying just the initial condition. However, in Figure 6.14, Dawes (2011) shows a general increase in peak height and shift to the right in peak position for Experiments 1 and 2, while this is not the case for the position of peak height of Experiment 3, in relation to initial conditions in Experiments 1 and 2. We choose not to compare to Figure 6.20 as this includes an additional parameter to the transition, which they find only to have a good fit for Experiment 2.

11. Section haded ‘EIP variation and identifiability’

From page 12 line 53 ‘EIP variation and identifiability’

Firstly, this section discusses correlations between various model parameters and output, and is not on the whole well-described by the section title.

As a general point the correlations discussed need to be better described in terms of the insights they offer regarding the biology being modelled, or else what they indicate of interest regarding any limitations of the model. At present the text here appears confused and is not telling a clear story about the meaning of the observed correlations.

See separate comment re relationship between f and μO .

We have reformulated and split this section into two. To better describe their contents, we have labeled them as “Parameter correlations” and “Parameter variability” and focused more on the biological interpretation. As there are more MCMC chains, we have included all the correlation plots in the GitHub Repository and placed the referenced correlation coefficients in a supplemental Table.

12. Figure 8.

(i) Why is the x axis for Figure 8 labelled 'time, days' when the axis labels are categorical?

The label "time, days" was an error and has been corrected to "Parameter"

(ii) Why is the 'N' parameter not included in the list of sensitised parameter values?

This sensitivity method only works for continuous variables, not discrete as is N. We are showing the results for the optimal N (found in Table 2). We also note the N used for each initial ookinete density in the caption.

13. Defined relationships between EIP and model parameters

Page 12 from line 38, and Page 13, from line 54. The authors define EIP as equal to the average time in the ookinete category plus the time taken for 50% of bursting oocysts to have burst. Stating that the parameters which primarily explain variation in the EIP are the parameters which specifically determine these two time periods is therefore surely merely stating the self-evident, isn't this an inevitable outcome of the definitions rather than an informative model result? If it does need to be stated, it definitely does not seem worth stating twice.

Indeed, the formula for EIP does not include all the parameters. However, the fitting process includes all the parameters, such that changes in parameters not specifically in the equation for EIP can lead to changes in the parameter values in the equation for EIP and thus alter the calculated EIP. Furthermore, it allows us to determine which parameters in the EIP equation are most sensitive.

14. Conclusion

Page 14, the section headed 'Conclusion' seems to state generalities, EIP is important and poorly understood, better understanding of EIP could support improved disease control. This states the context for the work presented in the *ms*, but offers no information about the work and results presented here.

The conclusion needs to provide a succinct summary of what this modelling exercise has achieved.

We have revised the conclusion section to include a summary of the important main results of our work.

References in Response to Reviewer

1. Sinden RE, Dawes EJ, Alavi Y, Waldock J, Finney O, Mendoza J, Butcher GA, Andrews L, Hill AV, Gilbert SC: **Progression of *Plasmodium berghei* through *Anopheles stephensi* is density-dependent.** *PLoS Pathogens* 2007, **3**:2005 - 2016.

2. Churcher TS, Dawes EJ, Sinden RE, Christophides GK, Koella JC, Basáñez M-G: **Population biology of malaria within the mosquito: density- dependent processes and potential implications for transmission-blocking interventions.** *Malaria Journal* 2010, **9**:311.
3. Dawes EJ, Zhuang S, Sinden RE, Basáñez M-G: **The temporal dynamics of Plasmodium density through the sporogonic cycle within Anopheles mosquitoes.** *Transactions of the Royal Society of Tropical Medicine and Hygiene* 2009, **103**:1197-1198.
4. Dawes E, Churcher T, Zhuang S, Sinden R, Basanez M-G: **Anopheles mortality is both age- and Plasmodium-density dependent: implications for malaria transmission.** *Malaria Journal* 2009, **8**:228.
5. Basáñez M-G FJ, Dawes EJ, Finney O, Sinden RE: **The impact of parasite density on life-stage transitions of Plasmodium within the mosquito vector.** *Proceedings of the 11th International Congress of Parasitology* 2006:11-18.
6. Paaijmans KP, Blanford S, Chan BHK, Thomas MB: **Warmer temperatures reduce the vectorial capacity of malaria mosquitoes.** *Biology Letters* 2011.
7. Shapiro LL, Murdock CC, Jacobs GR, Thomas RJ, Thomas MB: **Larval food quantity affects the capacity of adult mosquitoes to transmit human malaria.** *Proceedings of the Royal Society B: Biological Sciences* 2016, **283**:20160298.
8. Shapiro LLM, Whitehead SA, Thomas MB: **Quantifying the effects of temperature on mosquito and parasite traits that determine the transmission potential of human malaria.** *PLoS biology* 2017, **15**:e2003489.

FROM ATTACHED REVIEWER FILE

General Comments: (page numbers refer to those given in the header of the review proof, not those typeset by the journal template)

1. Several parts of the manuscript are rather too terse. This includes the methods section, where details about the experimental and quantitative analyses is incomplete to understand all analyses, but also the final discussion, and many of the figure and table captions. Specific comments are provided below to identify the sections that - in my opinion - would benefit from expansion.

We have significantly revised the methods and final discussion to improve readability. Individual changes are noted below.

2. I am not entirely convinced that all MCMC analyses ran to convergence. I would strongly suggest that the authors repeat their analyses using multiple MCMC chains with different starting values to ensure convergence was achieved. Similarly - although I am not an expert with multistart methods - I am not convinced that all of these analyses yielded sensible results. The model trajectories plotted in Fig. S3 for high values of the compartment number N shows curves for $E(0)=100,200,500$ and $N=75,100$ that are very poor fits to the data, when most other parameter combinations seem to yield reasonable fits. Is this a parameter identifiability issue for these particular situations, or is this a technical issue with the multistart procedure?

We thank the reviewer for the suggestion to revisit the convergence of our MCMC results, and in particular for the helpful guidance on how to appropriately do so.

In regards to the multistart procedure, we do find we have run to convergence. To confirm this, we have checked the exit flags from all multistart runs and achieve convergence, even for fits that look particularly bad, like the $E(0) = 200, N = 100$ case, or the $E(0) = 100, N = 100$ case. In these examples, for each of the 10 starting parameter sets, we get an exit flag of '2', which means that at least one local minimum was found for each of the 10 starting parameter sets, i.e. for each of the 10 starting points, *some* runs of the local solver converged. For reference, an exit flag of '1' also means that at least one local minimum was found, but in this case *all* local solvers converged. We found that most times we received an exit flag of 1 or 2 meaning that a local min was found at least once for each of the 10 initial parameter sets.

There are some parameter sets that yield an undefined negative log-likelihood. For these, we define the objective function value to be 10^9 . While this value leads to a positive exit flag, we omit these runs of multistart from our set of runs that converged. We have added Table to the supplemental information, which summarizes the convergence results for multistart. In particular, it indicates, for each initial ookinete density $E(0)$, and each model, how many of the 10 multistart runs converged (i.e., exit flag was 1 or 2, and the objective function value returned was less than 10^9). Of the 60 combinations, 1 resulted in 8/10 converging, 7 resulted in 9/10 converging, and the remaining 52 combinations resulted in 10/10 multistart runs converging.

For large N we expect the structure of the equations is inappropriate. To be clear, when N is

large, i.e. multiple transitions as an oocyst, results in an increasingly consistent timing to the ending of the oocyst stage and beginning of the sporozoite, which is not consistent with the data.

We have revised our methods section to discuss our convergence metrics. In addition, we have included supplemental tables with the Gelman Rubin diagnostics for our MCMC and summarizing our exit flags for the multistart runs. Furthermore, all the details of all runs can be found in our Github Repository: <https://github.com/laurenychilds/EIP>

3. In my opinion, the results indicate strong batch effects (i.e. parameter estimates clustering in Fig 6 for the two different experiments, as discussed on page 12 lines 23-27). My suspicion is that there may be an experimentally uncontrolled variable relating to e.g. the quality of the blood, the mosquito cohort, the accuracy with which $E(0)$ was known, or the quality/viability of the ookinetes that has a decided effect on the estimated parasite dynamics. Given this is a re-analysis of a thesis dataset, I appreciate that the authors may not be able to determine the underlying experimental factors for this, but this ought to be at least discussed in the manuscript as it may indicate that the level of replication in the experimental data is insufficient to capture the variability in the system (regardless whether it was caused by experimental/measurement error or natural variation).

We agree with the reviewer that we believe there are batch effects, particularly for f and μ_0 , and is the point we were trying to make in the text. We have enhanced this discussion to more clearly convey this message. Additionally, we point out that the y-axis for f and μ_0 are significantly narrower than the range of values considered for these parameters. If we were to plot on the full range, these batch effects would not be visually apparent.

Unfortunately, we do not have access to all details of the experiment to know why there would be batch effects. Two hypotheses discussed in the thesis (Dawes 2011) include: (i) difference in age of mosquitoes and (ii) difference in who performed counts. In experiment 1, the blood fed mosquitoes were aged 4-8 days while in experiments 2 and 3, they were aged 7-11 days (Dawes 2011, pg 112). We combined data for experiments 1 and 2 as the results were noted in the thesis that the data was so similar they wanted to change conditions for the third experiment (Dawes 2011, pg 110). The third experiment was counted by a masters student (Dawes 2011, pg 115) and they followed up and found no statistical significance in inter-experimenter variability (Dawes 2011, pg 118, Figure 4.5).

We add discussion of these points to the “Parameter Variability” section.

4. My understanding is that one of the key innovations of the manuscript is to model different classes of oocytes (bursting/non-bursting). Some parameters relating to the transitions between these classes appear to be only weakly identifiable from the available data (μ_e , σ_e , f). I believe the authors should relate this finding back to the experimental system, and perhaps highlight that more accurate inferences (potentially allowing to get at the unresolved question about the nature of the waiting-time process from oocyst formation to oocyst bursting) might be possible if the two (or more) classes could be experimentally differentiated in future studies.

We agree with the reviewer and have added this point to the discussion section.

5. I found the overall organisation of the presentation of results confusing and would suggest consolidating some of the tables and figures, and moving some material from the main text to the supplement and vice versa. I think that in particular it would be helpful to readers with a more experimental background if the raw data (Fig. S1) were presented in the main text. Additional suggestions are given in the specific comments.

We thank the reviewer for the suggestion and have moved some figures accordingly. We have chosen, however, to keep the experimental data in the supplement as it is a re-analysis of data that is not our own and can be found in the public domain. In addition, the data does appear in Figure 4 (former Figure 5) in the main text.

We have moved former Figure 2 (now Figure S2) to the supplement. We altered Figure 3 (former Figure 5) to include posterior intervals for the MCMC results. We have added a new Figure 4 (replacing former Figure 7), that looks at the timing and height of oocyst peaks, the timing when sporozoite curves rise and the height of sporozoite score.

We have removed former Figures S6-S11 from the manuscript as now there are 30 MCMC chains. Instead, we include a table with relevant correlation coefficients in the supplemental material. All correlation plots for all 30 MCMC chains are available in our Github Repository: <https://github.com/laurencchilds/EIP>

Specific comments:

6. Page 3, line 36: I think the fact that the experimental data started infections at the ookinete stage (and not by using blood containing gametocytes) should be restated in the final discussion to highlight where the experimental system may diverge from the natural cycle of the parasite.

We agree and have added this to the discussion.

7. Page 3, line 47: Please give full bibliographic details for the thesis (University, DOI or a comparable repository url if available). Also note, that a description of the data appears to have been published as Dawes et al. 2009 Transactions Royal Soc Trop Med Hygiene 103:1197-1198

The reviewer is indeed correct that a short description of the data was published in Dawes et al 2009. As noted to Reviewer 1, we have clarified in our manuscript that much of the experimental data presented in the Dawes thesis has been published, including a summary figure of some temporal oocyst data for the experiments conducted in Ch. 4 of the thesis (reference [3] above), and have cited these references accordingly.

8. Page 4: The data section was very difficult to understand. Given that the data are not formally published it would be helpful if more detail (where available) could be provided here.

The details of the data collection can be found in detail in the referenced thesis (Dawes 2011 Chapter 4). We have added to Section 2 on “Data” to more clearly describe the key points of the

experiments. We also more clearly reference where the full description of the data can be found (Chapter 4 of Dawes 2011).

8-1: line 10: give the total duration of the experiments (40 days according to Fig S1).

The experiments varied in length and in days where collection was performed with the longest experiment (Experiment 3) lasts up until day 42. We have noted in the “Data” section that the experiments varied in length with longest being 42 days.

8-2: line 10: give a sample size of mosquitoes dissected each day

We have added to the “Data” section that in each sampling 20 mosquitoes were dissected (Dawes 2011, pg 114).

8-3: line 12: clarify the term "score". My understanding is this is an "abundance score" is that correct?

Score refers to the magnitude of the abundance of sporozoites found, but it is not a precise count. We have edited the “Data” section to clarify this.

8-4: line 13: clarify what the control entailed. Parasite free blood? Is any data from control mosquitoes presented? If not, why not?

The control entailed mosquitoes fed directly on anesthetized *P. berghei* infected mice rather than infected blood with ookinetes. Further details on the control are not included and the control data is not presented in the thesis. Thus, we do not have access to any information related to this. To avoid confusion, as we are not using the control data in any way, we have now removed mentioning the control in the manuscript. However, we do still refer to the experiments as *controlled* as the conditions and initial conditions are highly controlled.

8-5: line 19: if the score is related to abundance, some sort of counting must have happened. I'd suggest to change this sentence to "counts of sporozoites were not recorded".

The sporozoites were not counted precisely, hence our choice of wording. According to the Dawes 2011 on pg 115:

“... The number of sporozoites located in the salivary glands was recorded using a ‘sporozoite score’ to give an indication of parasite number. Precise parasite enumeration was not possible due to the logistics of the intensive experimental schedule...”

We have revised our sentence in the “Data” Section to:

“Sporozoites were not directly counted; instead sporozoite abundance levels were assigned a score between 0 and 4 on a log scale.”

8-6: line 27: clarify "values" do you mean counts/abundances?

We mean that only sporozoite score, not sporozoite count are reported. We have revised this

sentence, as mentioned above, to clarify.

8-7: line 31: I think adding Fig S1 to the main text would greatly help in understanding the experiments.

We have chosen to keep the experimental data in the Supplemental material as it is a re-analysis of data that is not our own and can be found in the public domain. In addition, the data does appear in Figure 3 (former Figure 5) in the main text.

9. Page 4, lines 46-49: Referencing the appropriate panels of Fig S1 would not hurt.

This is true in all panels of Figure S1, but we now include references to which symbols we are discussing. Furthermore, we have lengthened the text to clarify the points.

10. Page 4, lines 54-57: This is a key motivation for your study then. State this in the penultimate paragraph of the introduction!

We have added this to the introduction as suggested. Furthermore, we have revised the referred to lines according to suggestions from Reviewer 1.

11. Figure 1: I would prefer a more verbose figure caption that explains the symbols for at least some of the different compartments without having to look them up in the text.

We expanded the caption to Figure 1 to include additional information on the compartments and parameters.

12. Table 1: if parameters p and m were not fitted, what value did you use for it in the simulations, and why?

We use $p*m = 60$ as the number of sporozoites per oocyst that successfully make it to the salivary glands has been reported as 54-72 (Sinden et al 2007, reference [1] to the previous Reviewer). As the parameters appear together they are not identifiable. We now discuss this at the beginning of the “Parameter Estimation” section.

13. Page 6, line 57: Isn't 10 raised to the power of the sporozoite scores? i.e. the scores is the exponent?

We have corrected this error.

14. Page 7, section (b): I believe the parameters that aren't estimated ought to be discussed in this section (see comment #12)

We now discuss the parameters we do not fit at the beginning of the “Parameter Estimation” section as mentioned above in response to #12.

15. Page 8 lines 6-10: I believe Figure S2 is an accidental duplication of another figure (fig 4?) -

as I am having trouble to align the text here and in the caption of Fig S2 with the actual figure.)

The reviewer is correct. We accidentally duplicated another figure. That has been corrected.

16. Page 8, section (i). The later results only seem to present AICc values, so why do you state that both AIC and AICc were used? Also, this section is devoid of references when there is a large body of literature about the AIC itself and model selection for dynamical models. Any reasons/justification why you chose this particular approach?

We have modified the text to clarify that we are using AICc. We have chosen AIC for model selection due to its simplicity and general applicability for model comparison. Due to our small sample size, however, we ultimately settled on the correction for small sample size, AICc. We have also added a reference to Burnham and Anderson's book on model selection:

Burnham, Kenneth P., and David R. Anderson. *Model Selection and Multimodel Inference A Practical Information-Theoretic Approach*. 2. ed. New York, NY: Springer New York, 2002.

17. Page 9, section (e): Please provide more details on the Bayesian analysis, in particular how MCMC convergence was assessed. Some parameter posteriors appear to either cover a very large proportion of the prior range (e.g. k for $e(0)=500$) and/or are concentrated at the prior bound. Please make sure that multiple chains that have different MCMC starting values do indeed converge to the same posterior. I appreciate that the deBInfer package does not provide the most efficient way of running multiple chains, but the posterior chains for sequential runs of with randomized starting values, can be extracted and joined up using e.g. the coda package to then calculate formal convergence diagnostics as outlined below:

```
#[set up the model and parameters]
#run multiple calls to de_mcmc, ensure that they have the same number of iterations, and the
same parameter set
chain1 <- de_mcmc(...)
chain2 <- de_mcmc(...)
chain3 <- de_mcmc(...)
#combine the samples into an mcmc.list object
all_chains <- coda::as.mcmc.list(list(chain1$samples, chain2$samples, chain3$samples))
#all_chains can now be manipulated with coda like any multi-chain analysis, e.g.
plot(all_chains)
gelman.diag(all_chains)
#To harness the samples from all chains in the deBInfer plotting functions and posterior
trajectory simulations etc. one can manipulate an existing debinfer_result object:
#NB: the chains will be merged using this approach
merged_results <- chain1 #make a copy of the output structure
merged_results$samples <- coda::as.mcmc(do.call(rbind, all_chains))
post_prior_densplot(merged_results, burnin = 1)
```

We thank the reviewer for the provided code. We have run five chains with parameters chosen using Latin Hypercube sampling across the parameter ranges to ensure that our starting points are diverse. Using the Gelman-Rubin convergence diagnostic, we determine all the chains have

converged. Our summary data now includes all five chains. Note: these new five chains do not include the original data but the results (medians, ranges and distributions) are nearly identical.

18. Table 2: I would consolidate this table into table 1 for conciseness.

We agree and now the parameter bounds are found in Table 1 and former Table 2 has been removed.

19. Table 3: I think it would be useful to add the credible intervals for each parameter for the Bayesian results. (And also use them in the subsequent discussion of how similar multistart and Bayesian results are, i.e. do the former fall within some posterior credible interval of the latter). I would also present the relevant AICc values in this table.

We agree that this information is important. However, Table 2 (former Table 3) is already very busy. We have instead chosen to add a supplemental table with the credible intervals for the parameters. We use this in our new discussion of the multistart and Bayesian results.

20. page 11. line 7: Give explicit AICc differences here. The y-axis scale in Figure 3 is so large (and variable among the different fits) that it is hard to see how similar the two different models performed exactly

As in #19, we have added a table in the supplement with the AICc values.

21. When discussing similarity of parameter estimates make use of the uncertainty estimates from the Bayesian fit (cf comment#19)

We revised our discussion of parameter variability to include the 95% HDPI. These values are found in a new Table in the supplement.

22. Page 4, line 18: Expand what you mean by "reasonable fit". Some of the fits in Fig S3, do indeed look not reasonable (see comment #2). Is this an issue of concern?

We are unsure of what the reviewer is referring to on Page 4. When we discuss "reasonable fit" in the results section, we note that we get similar parameters from all fits except when the fit seems to be unreasonable.

23. Page 11, line 31: Figure 7 is a bit crowded and the ordering of peaks is not easy to see. Can you add panels that plot the estimated time of the peak oocyst count / the estimated final sporozoite score against $E(0)$?

We have removed former Figure 7 and replaced with a figure (new Figure 4) that plots:

- Initial ookinete number vs timing of the peak in the oocyst number
- Initial ookinete number vs peak in the oocyst number
- Initial ookinete number vs timing of the rise of the sporozoite score
- Initial ookinete number vs height of the sporozoite score

24. The "wrong" ordering of the peaks may well be an experimental artefact caused by a batch effect of some sort (see comment #3). This should be expanded on here and/or in the discussion, especially as - as far as I understand - the dependence structure of the different experiments was not explicitly modelled (and probably can't be given the lack of replication).

We agree that we are unable to address batch effects from experiments explicitly in our model. However, only two of our parameters seem to indicate these batch effects, f and μ_O . Furthermore, the variation in these is very small, i.e. the parameters are very similar across all initial ookinete densities.

25. Page 11, lines 34-36: I am not clear about the point you are trying to make in the sentence starting "Thus, interestingly". Can you please clarify this sentence, and the link between f and the other estimated quantities?

We have revised this sentence for clarity.

26. page 12, first paragraph: I think there may be a parameter identifiability issue here that deserves mention, which is that you are trying to estimate transitions between latent classes of which you can only observe a sum count. Does this have implications for future experiments? (I think so. See comment #4)

We have revised our discussion of "Parameter Correlations" and "Parameter Variability" in the results section, creating a subsection for each. In particular, we have expanded our discussion of potential identifiability issues between parameters such as f and μ_O . Additionally, we have added discussion points about experiments to the discussion.

27. Line 17/Figure 6. Please ensure that the variation you describe is not an artefact of an uncovered MCMC!

We have confirmed that our MCMC have converged. We include these convergence diagnostics in a Table in the supplement.

28. Page 12, lines 23-27: Again, I believe that batch effects in the experiment are at work here.

We agree and have revised the text accordingly.

29. Page 13, Line 22: This is the place, where you should - in my opinion - tie the models back to the biological system and it's experimental treatment, and mention that further developments of the experimental approach are needed to check whether these classes of oocysts are indeed distinguishable in the real world.

We agree that additional experimental work could help distinguish parameters associated with ookinete stage and have noted this in the discussion. However, we do not expect it to be possible to distinguish stages of the oocysts class.

30. Page 13, line 30: I am not an expert on mosquito-plasmodium interactions, but besides

density dependent competition among the parasites couldn't there be also a dose-response relationship at play where the mosquito host reacts differently to different parasite concentrations?

We agree that there could be dose-responses occurring, such as via in mosquito immunity, with higher number of parasites. However, we would expect a monotonically increasing EIP in this case, not the U-shaped relationship with EIP we observe. Nevertheless, we have added this idea to the discussion.

32. Page 13, Line 47: Again, wouldn't it be nice if there was experimental data to allow a better separation of the estimates of μ_E and σ_E ? Do state this. Confronting models with data is a two-way street, and you should provide take-home messages for future experimentators on how they might ensure the collection of data that could better inform the models that you present.

We agree that additional experimental work could help distinguish parameters associated with ookinete stage and have noted this in the "Discussion" section.

33. Page 14, Section 7: See comment #32. I think you should put some take-home messages for future experiments here. Justify once more why you think understanding the waiting-time process matters biologically/epidemiologically and how that is related to your assumption of different oocyte classes, and then issue a call for experiments to identify these classes.

We agree that additional experimental work could help distinguish parameters associated with ookinete stage and have noted this in the discussion. However, we do not expect it to be possible to distinguish substages of the oocysts class. We have also re-iterated the importance of EIP in determining the likelihood of onward transmission from mosquitoes to humans in the first paragraph of the Discussion section.

34. Fig. 2: I would suggest moving this to the supplement.

We have done this.

35. Fig 5: Can you add posterior credible intervals around the presented MCMC draws?

Posterior credible intervals (as high density posterior intervals HDPI) were found in former Figure 7 in the original manuscript. We have removed former Figure 7, added HDPI to Figure 3, and added the figure suggested in comment #23.

36. Fig. 7: see comments #23

We have added the Figure suggested in comment #23 and removed this Figure.

37. Fig S1: I would suggest moving this into the main text.

As noted above, we have chosen to keep the experimental data in the Supplement as it is a re-analysis of data that is not our own and can be found in the public domain.

38. Fig S2: I think the figure shown here is not the correct figure, but a duplicate of Fig 4.

The reviewer is correct. We accidentally duplicated another figure. That has been corrected.

Appendix D

Royal Society
Open Access

July 20, 2020

Dear Editors,

We thank you for your helpful feedback, and for accepting our manuscript, "The impact of within-vector parasite development on the extrinsic incubation period", with minor revision. We have addressed the typographical errors and have responded to your request to include information about the comparison between different ways of quantifying the sporozoite data in the body of the manuscript, as well as detailed information in the supplemental document. You will find a summarized description of this comparison in the last paragraph of section 4(a), in which we discuss the likelihood function. We look forward to your response.

Best regards,

Olivia Prosper & Lauren Childs